# A Scalable Neural Network for DSIC Affine Maximizer Auction Design

**Zhijian Duan**
CFCS, School of Computer Science
Peking University
zjduan@pku.edu.cn

**Haoran Sun**
Peking University
sunhaoran0301@stu.pku.edu.cn

**Yurong Chen**
CFCS, School of Computer Science
Peking University
chenyurong@pku.edu.cn

**Xiaotie Deng**
CFCS, School of Computer Science
& CMAR, Institute for AI
Peking University
xiaotie@pku.edu.cn

## Abstract

Automated auction design aims to find empirically high-revenue mechanisms through machine learning. Existing works on multi item auction scenarios can be roughly divided into RegretNet-like and affine maximizer auctions (AMAs) approaches. However, the former cannot strictly ensure dominant strategy incentive compatibility (DSIC), while the latter faces scalability issue due to the large number of allocation candidates. To address these limitations, we propose AMenuNet, a scalable neural network that constructs the AMA parameters (even including the allocation menu) from bidder and item representations. AMenuNet is always DSIC and individually rational (IR) due to the properties of AMAs, and it enhances scalability by generating candidate allocations through a neural network. Additionally, AMenuNet is permutation equivariant, and its number of parameters is independent of auction scale. We conduct extensive experiments to demonstrate that AMenuNet outperforms strong baselines in both contextual and non-contextual multi-item auctions, scales well to larger auctions, generalizes well to different settings, and identifies useful deterministic allocations. Overall, our proposed approach offers an effective solution to automated DSIC auction design, with improved scalability and strong revenue performance in various settings.

## 1 Introduction

One of the central topics in auction design is to construct a mechanism that is both dominant strategy incentive compatible (DSIC) and individually rational (IR) while bringing high expected revenue to the auctioneer. The seminal work by Myerson [1981] characterizes the revenue-maximizing mechanism for single-parameter auctions. However, after four decades, the optimal auction design problem in multi-parameter scenarios remains incompletely understood, even in simple settings such as two bidders and two items [Dütting et al., 2019]. To solve the problem, recently, there has been significant progress in *automated auction design* [Sandholm and Likhodedov, 2015b, Dütting et al., 2019]. Such a paradigm formulates auction design as an optimization problem subject to DSIC and IR constraints and then finds optimal or near-optimal solutions using machine learning.

The works of automated auction design can be roughly divided into two categories. The first category is the RegretNet-like approach [Curry et al., 2020, Peri et al., 2021, Rahme et al., 2021b,a, Duan et al., 2022, Ivanov et al., 2022], pioneered by RegretNet [Dütting et al., 2019]. These works

37th Conference on Neural Information Processing Systems (NeurIPS 2023).

represent the auction mechanism as neural networks and then find near-optimal and approximate DSIC solutions using adversarial training. The second category is based on affine maximizer auctions (AMAs) [Roberts, 1979, Likhodedov and Sandholm, 2004, Likhodedov et al., 2005, Sandholm and Likhodedov, 2015a, Guo et al., 2017, Curry et al., 2023]. These methods restrict the auction mechanism to AMAs, which are inherently DSIC and IR. Afterward, they optimize AMA parameters using machine learning to achieve high revenue.

Generally, RegretNet-like approaches can achieve higher revenue than AMA-based approaches. However, these works are not DSIC. They can only ensure approximate DSIC by adding a regret term in the loss function as a penalty for violating the DSIC constraints. There are no theoretical results on the regret upper bound, and the impact of such regret on the behaviors of strategic bidders. Even worse, computing the regret term is time-consuming [Rahme et al., 2021b]. AMA-based approaches, on the other hand, offer the advantage of guaranteeing DSIC and IR due to the properties of AMAs. However, many of these approaches face scalability issues because they consider all deterministic allocations as the allocation menu. The size of the menu grows exponentially, reaching $(n+1)^m$ for $n$ bidders and $m$ items, making it difficult for these approaches to handle larger auctions. Even auctions with 3 bidders and 10 items can pose challenges to the AMA-based methods.

To overcome the limitations above, we propose a scalable neural network for the DSIC affine maximizer auction design. We refer to our approach as AMenuNet: Affine maximizer auctions with Menu Network. AMenuNet constructs the AMA parameters, including the allocation menu, bidder weights, and boost variables, from the bidder and item representations. After getting the parameters, we compute the allocation and payment results according to AMA. By setting the representations as the corresponding contexts or IDs, AMenuNet can handle both contextual [Duan et al., 2022] and non-contextual classic auctions. As AMenuNet only relies on the representations, the resulting mechanism is guaranteed to be DSIC and IR due to the properties of AMAs.

Specifically, we employ two techniques to address the scalability issue of AMA-based approaches: (1) Firstly, we parameterize the allocation menu. We predefine the size of the allocation menu and train the neural network to compute the allocation candidates within the menu, along with the bidder weights and boost variables. This allows for more efficient handling of large-scale auctions. (2) Secondly, we utilize a transformer-based permutation-equivariant architecture. Notably, this architecture's parameters remain independent of the number of bidders or items. This enhances the scalability of AMenuNet, enabling it to handle auctions of larger scales than those in training.

We conduct extensive experiments to demonstrate the effectiveness of AMenuNet. First, our performance experiments show that in both contextual and classic multi-item auctions, AMenuNet can achieve higher revenue than strong DSIC and IR baselines. AMenuNet can also achieve comparable revenue to RegretNet-like approaches that can only ensure approximate DSIC. Next, our ablation study shows that the learnable allocation menu provides significant benefits to AMenuNet, from both revenue and scalability perspectives. Thirdly, we find that AMenuNet can also generalize well to auctions with a different number of bidders or items than those in the training data. And finally, the case study of winning allocations illustrates that AMenuNet can discover useful deterministic allocations and set the reserve price for the auctioneer.

## 2   Related Work

Automated mechanism design[Conitzer and Sandholm, 2002, 2004, Sandholm and Likhodedov, 2015b] has been proposed to find approximate optimal auctions with multiple items and bidders [Balcan et al., 2008, Lahaie, 2011, Dütting et al., 2015]. Meanwhile, several works have analyzed the sample complexity of optimal auction design problems [Cole and Roughgarden, 2014, Devanur et al., 2016, Balcan et al., 2016, Guo et al., 2019, Gonczarowski and Weinberg, 2021]. Recently, pioneered by RegretNet [Dütting et al., 2019], there is rapid progress on finding (approximate) optimal auction through deep learning [Curry et al., 2020, Peri et al., 2021, Rahme et al., 2021b,a, Duan et al., 2022, Ivanov et al., 2022]. However, as we mentioned before, those RegretNet-like approaches can only ensure approximate DSIC by adding a hard-to-compute regret term.

Our paper follows the affine maximizer auction (AMA) [Roberts, 1979] based approaches. AMA is a weighted version of VCG [Vickrey, 1961], which assigns weights $w$ to each bidder and assigns boosts to each feasible allocation. Tuning the weights and boosts enables AMAs to achieve higher revenue than VCG while maintaining DSIC and IR. Different subsets of AMA have been studied

in various works, such as VVCA [Likhodedov and Sandholm, 2004, Likhodedov et al., 2005, Sandholm and Likhodedov, 2015b], $\lambda$-auction [Jehiel et al., 2007], mixed bundling auction [Tang and Sandholm, 2012], and bundling boosted auction [Balcan et al., 2021]. However, these works set all the deterministic allocations as candidates, the size of which grows exponentially with respect to the auction scale. To overcome such issue, we construct a neural network that automatically computes the allocation menu from the representations of bidders and items. Curry et al. [2023] is the closest work to our approach that also parameterizes the allocation menu. The key difference is that they optimize the AMA parameters explicitly, while we derive the AMA parameters by a neural network and optimize the network weights instead. By utilizing a neural network, we can handle contextual auctions by incorporating representations as inputs. Additionally, the trained model can generalize to auctions of different scales than those encountered during training.

Contextual auctions are a more general and realistic auction format assuming that every bidder and item has some public information. They have been widely used in industry [Zhang et al., 2021, Liu et al., 2021]. In the academic community, previous works on contextual auctions mostly focus on the online setting of some well-known contextual repeated auctions, such as posted-price auctions [Amin et al., 2014, Mao et al., 2018, Drutsa, 2020, Zhiyanov and Drutsa, 2020], where the seller prices the item to sell to a strategic buyer, or repeated second-price auctions [Golrezaei et al., 2021]. In contrast, our paper focuses on the offline setting of contextual sealed-bid auctions, similar to Duan et al. [2022], a RegretNet-like approach.

## 3 Preliminary

**Sealed-Bid Auction.** We consider a sealed-bid auction with $n$ bidders and $m$ items. Denote $[n] = \{1, 2, \ldots, n\}$. Each bidder $i \in [n]$ is represented by a $d_x$-dimensional vector $\boldsymbol{x}_i \in \mathbb{R}^{d_x}$, which can encode her unique ID or her context (public feature) [Duan et al., 2022]. Similarly, each item $j \in [m]$ is represented by a $d_y$-dimensional vector $\boldsymbol{y}_j \in \mathbb{R}^{d_y}$, which can also encode its unique ID or context. By using such representations, we unify the contextual auction and the classical Bayesian auction. We denote by $X = [\boldsymbol{x}_1, \boldsymbol{x}_2, \ldots, \boldsymbol{x}_n]^T \in \mathbb{R}^{n \times d_x}$ and $Y = [\boldsymbol{y}_1, \boldsymbol{y}_2, \ldots, \boldsymbol{y}_m]^T \in \mathbb{R}^{m \times d_y}$ the matrices of bidder and item representations, respectively. These matrices follow underlying joint probability distribution $F_{X,Y}$. In an additive valuation setting, each bidder $i$ values each bundle of items $S \subseteq [m]$ with a valuation $v_{i,S} = \sum_{j \in S} v_{ij}$. The bidder has to submit her bids for each item $j \in [m]$ as $\boldsymbol{b}_i := (b_1, b_2, \ldots, b_m)$. The valuation profile $V = (v_{ij})_{i \in [n], j \in [m]} \in \mathbb{R}^{n \times m}$ is generated from a conditional distribution $F_{V|X,Y}$ that depends on the representations of bidders and items. The auctioneer does not know the true valuation profile $V$ but can observe the public bidder representations $X$, item representations $Y$, and the bidding profile $B = (b_{ij})_{i \in [n], j \in [m]} \in \mathbb{R}^{n \times m}$.

**Auction Mechanism.** An auction mechanism $(g, p)$ consists of an allocation rule $g : \mathbb{R}^{n \times m} \times \mathbb{R}^{n \times d_x} \times \mathbb{R}^{m \times d_y} \to [0, 1]^{n \times m}$ and a payment rule $p : \mathbb{R}^{n \times m} \times \mathbb{R}^{n \times d_x} \times \mathbb{R}^{m \times d_y} \to \mathbb{R}^n_{\geq 0}$. Given the bids $B$, bidder representations $X$, and item representations $Y$, $g_{ij}(B, X, Y) \in [0, 1]$ computes the probability that item $j$ is allocated to bidder $i$. We require that $\sum_{i=1}^{n} g_{ij}(B, X, Y) \leq 1$ for any item $j$ to guarantee that no item is allocated more than once. The payment $p_i(B, X, Y) \geq 0$ computes the price that bidder $i$ needs to pay. Bidders aim to maximize their own utilities. In the additive valuation setting, bidder $i$'s utility is $u_i(\boldsymbol{v}_i, B; X, Y) := \sum_{j=1}^{m} g_{ij}(B, X, Y)v_{ij} - p_i(B, X, Y)$, given $\boldsymbol{v}, B, X, Y$. Bidders may misreport their valuations to benefit themselves. Such strategic behavior among bidders could make the auction result hard to predict. Therefore, we require the auction mechanism to be *dominant strategy incentive compatible* (DSIC), which means that for each bidder $i \in [n]$, reporting her true valuation is her optimal strategy regardless of how others report. Formally, let $B_{-i} = (\boldsymbol{b}_1, \ldots, \boldsymbol{b}_{i-1}, \boldsymbol{b}_{i+1}, \ldots, \boldsymbol{b}_n)$ be the bids except for bidder $i$. A DSIC mechanism satisfies

$$u_i(\boldsymbol{v}_i, (\boldsymbol{v}_i, B_{-i}); X, Y)) \geq u_i(\boldsymbol{v}_i, (\boldsymbol{b}_i, B_{-i}); X, Y)), \quad \forall i, \boldsymbol{v}_i, B_{-i}, X, Y, \boldsymbol{b}_i. \tag{DSIC}$$

Furthermore, the auction mechanism needs to be *individually rational* (IR), which ensures that truthful bidding results in a non-negative utility for each bidder. Formally,

$$u_i(\boldsymbol{v}_i, (\boldsymbol{v}_i, B_{-i}); X, Y) \geq 0, \quad \forall i, \boldsymbol{v}_i, B_{-i}, X, Y. \tag{IR}$$

**Affine Maximizer Auction (AMA).** AMA [Roberts, 1979] is a generalized version of VickreyClarkeGroves (VCG) auction [Vickrey, 1961] that is inherently DISC and IR. An AMA consists

of positive weights $w_i \in \mathbb{R}_+$ for each bidder and boost variables $\lambda(A) \in \mathbb{R}$ for each allocation $A \in \mathcal{A}$, where $\mathcal{A}$ is the *allocation menu*, i.e., the set of all the feasible (possibly random) allocations. Given $B, X, Y$, AMA chooses the allocation that maximizes the affine welfare:

$$g(B, X, Y) = A^* := \arg\max_{A \in \mathcal{A}} \sum_{i=1}^{n} w_i b_i(A) + \lambda(A), \qquad \text{(Allocation)}$$

where $b_i(A) = \sum_{j=1}^{m} b_{ij} A_{ij}$ in additive valuation setting. The payment for bidder $k$ is

$$p_k(B, X, Y) = \frac{1}{w_k}\left(\sum_{i \neq k} w_i b_i(A^*_{-k}) + \lambda(A^*_{-k})\right) - \frac{1}{w_k}\left(\sum_{i \neq k} w_i b_i(A^*) + \lambda(A^*)\right). \quad \text{(Payment)}$$

Here we denote by $A^*_{-k} := \arg\max_{A \in \mathcal{A}} \sum_{i \neq k} w_i b_i(A) + \lambda(A)$ the allocation that maximizes the affine welfare, with bidder $k$'s utility excluded. Our definition of AMA differs slightly from previous literature [Roberts, 1979], whose allocation candidates are all the $(n+1)^m$ deterministic solutions. Instead, we explicitly define the allocation menu so that our definition is more general. As we will show in Appendix A, such AMAs are still DSIC and IR.

## 4 Methodology

In this section, we introduce the optimization problem for automated mechanism design and our proposed approach, AMenuNet, along with its training procedure.

### 4.1 Auction Design as an Optimization Problem

We begin by parameterizing our auction mechanism as $(g^\theta, p^\theta)$, where $\theta$ represents the neural network weights to be optimized. We denote the parameter space as $\Theta \ni \theta$ and the class of all parameterized mechanisms as $\mathcal{M}^\Theta$. The optimal auction design seeks to find the revenue-maximizing mechanism that satisfies both DSIC and IR:

$$\max_{\theta \in \Theta} \quad \text{Rev}(g^\theta, p^\theta) := \mathbb{E}_{(V,X,Y) \sim F_{V,X,Y}}\left[\sum_{i=1}^{n} p_i^\theta(V, X, Y)\right] \qquad \text{(OPT)}$$
$$\text{s.t.} \quad (g^\theta, p^\theta) \text{ is DSIC and IR}$$

where $\text{Rev}(g^\theta, p^\theta)$ is the expected revenue of mechanism $(g^\theta, p^\theta)$. However, there is no known characterization of a DSIC general multi-item auction [Dütting et al., 2019]. Thus, we restrict the search space of auctions to affine maximizer auctions (AMAs) [Roberts, 1979]. AMAs are inherently DSIC and IR, and can cover a broad class of mechanisms: Lavi et al. [2003] has shown that every DSIC multi-item auction (where each bidder only cares about what they get and pay) is almost an AMA (with some qualifications).

The AMA parameters consist of positive weights $\boldsymbol{w} \in \mathbb{R}_+^n$ for all bidders, an allocation menu $\mathcal{A}$, and boost variables $\boldsymbol{\lambda} \in \mathbb{R}^{|\mathcal{A}|}$ for each allocation in $\mathcal{A}$. While the most straightforward way to define $\mathcal{A}$ is to include all deterministic allocations, as done in prior work [Likhodedov and Sandholm, 2004, Likhodedov et al., 2005, Sandholm and Likhodedov, 2015b], this approach suffers from scalability issues as the number of allocations can be as large as $(n+1)^m$. To address this challenge, we propose AMenuNet, which predefines the size of $\mathcal{A}$ and constructs it along with the other AMA parameters using a permutation-equivariant neural network.

### 4.2 AMenuNet Architecture

Denote by $s = |\mathcal{A}|$ the predefined size of the allocation menu. AMenuNet takes as input the representations of bidders $X$ and items $Y$, and constructs all the $s$ allocations, the bidder weights $\boldsymbol{w}$, and the boost variables $\boldsymbol{\lambda} \in \mathbb{R}^s$ through a permutation-equivariant neural network architecture as illustrated in Figure 1. The architecture consists of an encode layer, a menu layer, a weight layer, and a boost layer. The final allocation and payment results can be computed by combining the submitted bids $B$ according to AMA (see Equation (Allocation) and Equation (Payment)).

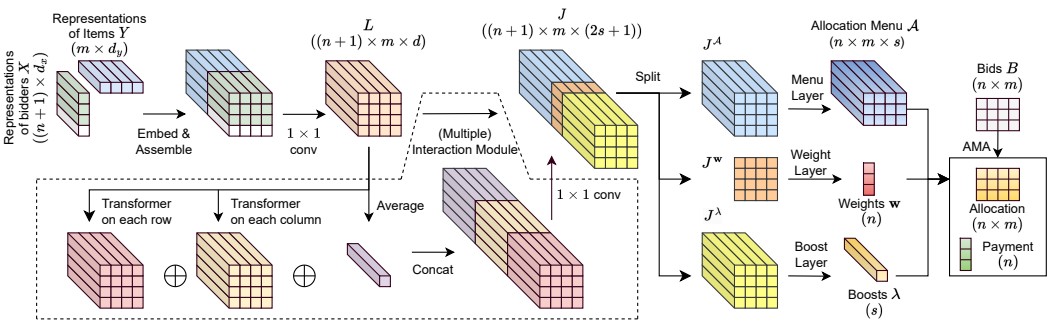

Figure 1: A schematic view of AMenuNet, which takes the bidder representations $X$ (including the dummy bidder) and item representations $Y$ as inputs. These representations are assembled into a tensor $E \in \mathbb{R}^{(n+1) \times m \times (d_x + d_y)}$. Two $1 \times 1$ convolution layers are then applied to obtain the tensor $L$. Following $L$, multiple transformer-based interaction modules are used to model the mutual interactions among all bidders and items. The output tensor after these modules is denoted as $J$. $J$ is further split into three parts: $J^{\mathcal{A}} \in \mathbb{R}^{(n+1) \times m \times s}$, $J^{\boldsymbol{w}} \in \mathbb{R}^{(n+1) \times m}$ and $J^{\boldsymbol{\lambda}} \in \mathbb{R}^{(n+1) \times m \times s}$. These parts correspond to the allocation menu $\mathcal{A}$, bidder weights $\boldsymbol{w}$, and boosts $\boldsymbol{\lambda}$, respectively. Finally, based on the induced AMA parameters and the submitted bids, the allocation and payment results are computed according to the AMA mechanism.

We first introduce a dummy bidder $n + 1$ to handle cases where no items are allocated to any of the $n$ bidders. We set such dummy bidder's representation $\boldsymbol{x}_{n+1}$ as a $d_x$-dimensional vector with all elements set to 1.

**Encode Layer.** The encode layer transforms the initial representations of all bidders (including the dummy bidder) and items into a joint representation that captures their mutual interactions. In contextual auctions, their initial representations are the contexts. In non-contextual auctions, similar to word embedding [Mikolov et al., 2013], we embed the unique ID of each non-dummy bidder or item into a continuous vector space, and use the embeddings as their initial representations. Based on that, we construct the initial encoded representations for all pairs of $n + 1$ bidders and $m$ items $E \in \mathbb{R}^{(n+1) \times m \times (d_x + d_y)}$, where

$$E_{ij} = [\boldsymbol{x}_i; \boldsymbol{y}_j] \in \mathbb{R}^{d_x + d_y}$$

is the initial encoded representation of bidder $i$ and item $j$.

We further model the mutual interactions of bidders and items by three steps. Firstly, we capture the inner influence of each bidder-item pair by two $1 \times 1$ convolutions with a ReLU activation. By doing so, we get

$$L = \text{Conv}_2 \circ \text{ReLU} \circ \text{Conv}_1 \circ E \in \mathbb{R}^{(n+1) \times m \times d},$$

where $d$ is the dimension of the new latent representation for each bidder-item pair, both $\text{Conv}_1$ and $\text{Conv}_2$ are $1 \times 1$ convolutions, and $\text{ReLU}(x) := \max(x, 0)$.

Secondly, we model the mutual interactions between all the bidders and items by using the transformer [Vaswani et al., 2017] based interaction module, similar to Duan et al. [2022]. Specifically, for each bidder $i$, we model her interactions with all the $m$ items through transformer on the $i$-th row of $L$, and for each item $j$, we model its interactions with all the $n$ bidders through another transformer on the $j$-th column of $L$:

$$I_{i,*}^{\text{row}} = \text{transformer}(L_{i,*}) \in \mathbb{R}^{m \times d_h}, \quad I_{*,j}^{\text{column}} = \text{transformer}(L_{*,j}) \in \mathbb{R}^{(n+1) \times d_h},$$

where $d_h$ is the size of the hidden nodes of transformer; For all the bidders and items, their global representation is obtained by the average of all the representations

$$e^{\text{global}} = \frac{1}{(n+1)m} \sum_{i=1}^{n+1} \sum_{j=1}^{m} L_{ij}.$$

Thirdly, we get the unified interacted representation

$$I_{ij} := [I_{ij}^{\text{row}}; I_{ij}^{\text{column}}; e^{\text{global}}] \in \mathbb{R}^{2d_h + d}$$

by combining all the three representations for bidder $i$ and item $j$. Two $1 \times 1$ convolutions with a ReLU activation are applied on $I$ to encode $I$ into the joint representation

$$I^{\text{out}} := \text{Conv}_4 \circ \text{ReLU} \circ \text{Conv}_3 \circ I \in \mathbb{R}^{(n+1) \times m \times d_{\text{out}}}.$$

By stacking multiple interaction modules, we can model higher-order interactions among all bidders and items.

For the final interaction module, we set $d_{\text{out}} = 2s + 1$ and we denote by $J \in \mathbb{R}^{(n+1) \times m \times (2s+1)}$ its output. We then partition $J$ into three tensors: $J^{\mathcal{A}} \in \mathbb{R}^{(n+1) \times m \times s}$, $J^{\boldsymbol{w}} \in \mathbb{R}^{(n+1) \times m}$ and $J^{\boldsymbol{\lambda}} \in \mathbb{R}^{(n+1) \times m \times s}$, and pass them to the following layers.

**Menu Layer.** To construct the allocation menu, we first normalize each column of $J^{\mathcal{A}}$ through softmax function. Specifically, $\forall j \in [m], k \in [s]$, we denote by $J^{\mathcal{A}}_{*,j,k} \in \mathbb{R}^{n+1}$ the $j$-th column of allocation option $k$, and obtain its normalized probability $\tilde{J}^{\mathcal{A}}_{*,j,k}$ by

$$\tilde{J}^{\mathcal{A}}_{*,j,k} := \text{Softmax}(\tau \cdot J^{\mathcal{A}}_{*,j,k})$$

where $\text{Softmax}(\boldsymbol{x})_i := e^{x_i} / (\sum_{k=1}^{n+1} e^{x_k}) \in (0,1)$ for all $\boldsymbol{x} \in \mathbb{R}^{n+1}$ is the softmax function, and $\tau > 0$ is the temperature parameter. Next, we exclude the probabilities associated with the $n+1$-th dummy bidder to obtain the allocation menu:

$$\mathcal{A} = \tilde{J}^{\mathcal{A}}_{1:n} \in \mathbb{R}^{n \times m \times s}.$$

Thus, for each allocation $A \in \mathcal{A}$ we satisfy $A_{ij} \in [0,1]$ and $\sum_{i=1}^{n} A_{ij} \leq 1$ for any bidder $i$ and any item $j$.

**Weight Layer.** In this work, we impose the constraint that each bidder weight lies in $(0,1)$. Given $J^{\boldsymbol{w}} \in \mathbb{R}^{(n+1) \times m}$, for each bidder $i \leq n$ we get her weight by

$$w_i = \text{Sigmoid}\left(\frac{1}{m} \sum_{j=1}^{m} J^{\boldsymbol{w}}_{ij}\right),$$

where $\text{Sigmoid}(x) := 1/(1 + e^{-x}) \in (0,1)$ for all $x \in \mathbb{R}$ is the sigmoid function.

**Boost layer.** In boost layer, we first average $J^{\boldsymbol{\lambda}}$ across all bidders and items. We then use a multi-layer perceptron (MLP), which is a fully connected neural network, to get the boost variables $\boldsymbol{\lambda}$. Specifically, we compute $\boldsymbol{\lambda}$ by

$$\boldsymbol{\lambda} = \text{MLP}\left(\sum_{i=1}^{n+1} \sum_{j=1}^{m} J^{\boldsymbol{\lambda}}_{ij}\right) \in \mathbb{R}^s.$$

After we get the output AMA parameters $\mathcal{A}, \boldsymbol{w}, \boldsymbol{\lambda}$ from AMenuNet, we can compute the allocation result according to Equation (Allocation) and the payment result according to Equation (Payment). The computation of $\mathcal{A}, \boldsymbol{w}, \boldsymbol{\lambda}$ only involves the public representations of bidders and items, without access to the submitted bids. Therefore, the mechanism is DSIC and IR (see Appendix A for proof):

**Theorem 4.1.** *The mechanism induced by AMenuNet satisfies both DSIC and IR.*

Moreover, As AMenuNet is built using equivariant operators such as transformers and $1 \times 1$ convolutions, any permutation of the inputs to AMenuNet, including the submitted bids $B$, bidder representations $X$, and item representations $Y$, results in the same permutation of the allocation and payment outcomes. This property is known as permutation equivariance [Rahme et al., 2021a, Duan et al., 2023]:

**Definition 4.2** (Permutation Equivariance)**.** We say $(g, p)$ is permutation equivariant, if for any $B, X, Y$ and for any two permutation matrices $\Pi_n \in \{0,1\}^{n \times n}$ and $\Pi_m \in \{0,1\}^{m \times m}$, we have $g(\Pi_n B \Pi_m, \Pi_n X, \Pi_m^T Y) = \Pi_n g(B, X, Y) \Pi_m$ and $p(\Pi_n B \Pi_m, \Pi_n X, \Pi_m^T Y) = \Pi_n p(B, X, Y)$.

Permutation equivariant architectures are widely used in automated auction design [Rahme et al., 2021a, Duan et al., 2022, Ivanov et al., 2022]. Qin et al. [2022] have shown that this property can lead to better generalization ability of the mechanism model.

### 4.3 Optimization and Training

Since the induced AMA is DSIC and IR, we only need to maximize the expected revenue $\text{Rev}(g^\theta, p^\theta)$. To achieve this, following the standard machine learning paradigm [Shalev-Shwartz and Ben-David, 2014], we minimize the negative empirical revenue by set the loss function as

$$\ell(\theta, S) := \frac{1}{|S|} \sum_{k=1}^{|S|} \sum_{i=1}^{n} -p_i^\theta(V^{(k)}, X^{(k)}, Y^{(k)}),$$

where $S$ is the training data set, and $\theta$ contains all the neural network weights in the encode layer and the boost layer. However, the computation of $\ell(\theta, S)$ involves finding the affine welfare-maximizing allocation scheme $A^*$ (and $A^*_{-k}$), which is non-differentiable. To address this challenge, we use the softmax function as an approximation. During training, we compute an approximate $A^*$ by

$$\widetilde{A^*} := \frac{1}{Z} \sum_{A \in \mathcal{A}} \exp\left(\tau_A \cdot \left(\sum_{i=1}^{n} w_i b_i(A) + \boldsymbol{\lambda}(A)\right)\right) \cdot A,$$

where $Z := \sum_{A' \in \mathcal{A}} \exp\left(\tau_A \cdot \left(\sum_{i=1}^{n} w_i b_i(A') + \boldsymbol{\lambda}(A')\right)\right)$ is the normalizer, and $\tau_A$ is the softmax temperature. We can control the approximation level of $\widetilde{A^*}$ by tuning $\tau_A$: when $\tau_A \to \infty$, $\widetilde{A^*}$ recover the true $A^*$, and when $\tau_A \to 0$, $\widetilde{A^*}$ tends to a uniform combination of all the allocations in $\mathcal{A}$. The approximation $\widetilde{A^*_{-k}}$ of $A^*_{-k}$ is similar. By approximating $A^*$ and $A^*_{-k}$ through the differentiable $\widetilde{A^*}$ and $\widetilde{A^*_{-k}}$, we make it feasible to optimize $\ell(\theta, S)$ through gradient descent. Notice that in testing, we still follow the standard computation in Equation (Allocation) and Equation (Payment).

## 5 Experiments

In this section, we present empirical experiments that evaluate the effectiveness of AMenuNet[1]. All experiments are run on a Linux machine with NVIDIA Graphics Processing Unit (GPU) cores. Each result is obtained by averaging across 5 different runs. In all experiments, the standard deviation of AMenuNet across different runs is less than 0.01.

**Baseline Methods.** We compare AMenuNet against the following baselines:

1. VCG [Vickrey, 1961], which is the most classical special case of AMA;
2. Item-Myerson, a strong baseline used in Dütting et al. [2019], which independently applies Myerson auction with respect to each item;
3. Lottery AMA [Curry et al., 2023], an AMA-based approach that directly sets the allocation menu, bidder weights, and boost variables as all the learnable weights.
4. RegretNet [Dütting et al., 2019], the pioneer work of applying deep learning in auction design, which adopts fully-connected neural networks to compute auction mechanisms.
5. CITransNet [Duan et al., 2022], a RegretNet-like transformer-based approach that supports contextual auctions.

Note that both RegretNet and CITransNet can only achieve approximate DSIC by adding a regret term in the loss function. We train both models with small regret, less than 0.005.

**Hyperparameters.** We train the models for a maximum of 8000 iterations, with 32768 generated samples per iteration. The batch size is 2048, and we evaluate all models on 100000 samples. We set the softmax temperature as 500 and the learning rate as $3 \times 10^{-4}$. We tune the menu size in $\{32, 64, 128, 256, 512, 1024\}$[2]. For the boost layer, we use a two-layer fully connected neural network with ReLU activation. Given the induced AMA parameters, our implementation of the remaining AMA mechanism is built upon the implementation of Curry et al. [2023]. Further implementation details can be found in Appendix B.

---

[1]Our implementation is available at `https://github.com/Haoran0301/AMenuNet`

[2]We explore the impact of different menu sizes in Appendix C.

Table 1: The experiment results of average revenue. For each case, we use the notation $n \times m$ to represent the number of bidders $n$ and the number of items $m$. The regret of both CITransNet and RegretNet is less than 0.005. The best revenue is highlighted in **bold**, and the best revenue among all DSIC methods is underlined.

(a) Contextual auctions (Setting (A) and (B)). We omit Lottery AMA and RegretNet since they are designed for classic auctions. "Randomized Allocations" shows the ratio of randomized allocations output by AMenuNet.

| Method | DSIC? | $2\times2$ (A) | $2\times5$ (A) | $2\times10$ (A) | $3\times2$ (A) | $3\times5$ (A) | $3\times10$ (A) | $4\times2$ (B) | $5\times2$ (B) | $6\times2$ (B) | $7\times2$ (B) |
|---|---|---|---|---|---|---|---|---|---|---|---|
| CITransNet | No | **0.4461** | **1.1770** | **2.4218** | 0.5576 | **1.4666** | **2.9180** | **0.7227** | **0.7806** | **0.8396** | **0.8997** |
| VCG | Yes | 0.2882 | 0.7221 | 1.4423 | 0.4582 | 1.1457 | 2.2967 | 0.5751 | 0.6638 | 0.7332 | 0.7904 |
| Item-Myerson | Yes | 0.4265 | 1.0700 | 2.1398 | 0.5590 | 1.3964 | 2.7946 | 0.6584 | 0.7367 | 0.8004 | 0.8535 |
| AMenuNet | Yes | 0.4398 | 1.1539 | 2.3935 | **0.5601** | 1.4458 | 2.9006 | 0.7009 | 0.7548 | 0.8127 | 0.8560 |
| Randomized Allocations | | 1.42% | 7.37% | 18.01% | 3.52% | 22.25% | 54.38% | 12.41% | 14.81% | 10.76% | 17.02% |

(b) Classic auctions (Setting (C)-(F)).

| Method | DSIC? | $2\times5$(C) | $3\times10$(C) | $5\times5$(C) | $3\times1$(D) | $1\times2$(E) | $1\times2$(F) |
|---|---|---|---|---|---|---|---|
| Optimal | - | - | - | - | 2.7490 | **9.7810** | 0.1706 |
| CITransNet | No | **2.3788** | **5.9191** | **3.4759** | **2.7541** | 9.7551 | 0.1691 |
| RegretNet | No | 2.3390 | 5.5410 | 3.4260 | 2.7264 | 9.7340 | **0.1732** |
| VCG | Yes | 1.6645 | 5.0002 | 3.3319 | 2.4954 | 0 | 0 |
| Item-Myerson | Yes | 2.0755 | 5.3141 | 3.3566 | 2.7490 | 8.8359 | 0.1482 |
| Lottery AMA | Yes | 2.2354 | 5.3450 | 2.8527 | 2.7238 | 9.3139 | 0.1544 |
| AMenuNet | Yes | 2.2768 | 5.5896 | 3.3916 | 2.7382 | 9.6219 | 0.1701 |

**Auction Settings.** AMenuNet can deal with both contextual auctions and classic Bayesian auctions. We construct the following multi-item contextual auctions:

(A) We generate each bidder representations $\boldsymbol{x}_i \in \mathbb{R}^{10}$ and item representations $\boldsymbol{y}_j \in \mathbb{R}^{10}$ independently from a uniform distribution in $[-1,1]^{10}$ (i.e., $U[-1,1]^{10}$). The valuation $v_{ij}$ is sampled from $U[0, \text{Sigmoid}(\boldsymbol{x}_i^T \boldsymbol{y}_j)]$. This contextual setting is also used in Duan et al. [2022]. We choose the number of bidders $n \in \{2,3\}$ and the number of items $m \in \{2,5,10\}$.

(B) We set 2 items, and the representations are generated from the same way as in Setting (A). For valuations, we first generate an auxiliary variable $v_i'$ from $U[0,1]$ for each bidder $i$, and then we set $v_{i1} = v_i' \cdot \text{Sigmoid}(\boldsymbol{x}_i^T \boldsymbol{y}_1)$ and $v_{i2} = (1 - v_i') \cdot \text{Sigmoid}(\boldsymbol{x}_i^T \boldsymbol{y}_2)$. We do this to make the valuations of the 2 items highly correlated. We choose the number of bidders $n \in \{4,5,6,7\}$.

For classic auctions, we can assign each bidder and item a unique ID and embed them into a multi-dimensional continuous representation. We construct the following classic auction settings:

(C) For all bidders and items, $v_{ij} \sim U[0,1]$. Such setting is widely evaluated in RegretNet-like approaches [Dütting et al., 2019]. We select $n \times m$ (the number of bidders and items) in $\{2 \times 5, 3 \times 10, 5 \times 5\}$.

(D) 3 bidders and 1 item, with $v_{i1} \sim \text{Exp}(3)$ (i.e. the density function is $f(x) = 1/3e^{-1/3x}$ for all $i \in \{1,2,3\}$). The optimal solution is given by Myerson auction.

(E) 1 bidder and 2 items, with $v_{11} \sim U[4,7]$ and $v_{12} \sim U[4,16]$. The optimal auction is given by Daskalakis et al. [2015].

(F) 1 bidder and 2 items, where $v_{11}$ has the density function $f(x) = 5/(1+x)^6$, and $v_{12}$ has the density function $f(x) = 6/(1+x)^7$. The optimal solution is given by Daskalakis et al. [2015].

**Revenue Experiments.** The results of revenue experiments are presented in Table 1. We can see that in settings with unknown optimal solutions, AMenuNet achieves the highest revenue among all the DSIC approaches. The comparison between AMenuNet and VCG reveals that incorporating affine parameters into VCG leads to higher revenue. Notably, AMenuNet even surpasses the strong baseline Item-Myerson, which relies on prior distribution knowledge, an over strong assumption that may not hold especially in contextual auctions. In contrast, AMenuNet constructs the mechanism solely from sampled data, highlighting its data efficiency. Furthermore, AMenuNet demonstrates its

Table 2: The revenue results of ablation study. For each case, we use the notation $n \times m$ to represent the number of bidders $n$ and the number of items $m$. Some results of $\mathcal{A}_{\mathrm{dtm}}$ are intractable because the menu size is too large.

|  | 2×2(A) | 2×10(A) | 3×2(B) | 5×2(B) | 2×3(C) | 3×10(C) |
|---|---|---|---|---|---|---|
| AMenuNet | **0.4293** | 2.3815 | **0.6197** | **0.7548** | **1.3107** | **5.5896** |
| With $w = 1$ | 0.4209 | **2.3848** | 0.6141 | 0.7010 | 1.2784 | 5.5873 |
| With $\lambda = 0$ | 0.3701 | 2.2229 | 0.3892 | 0.3363 | 1.2140 | 5.4145 |
| With $\mathcal{A}_{\mathrm{dtm}}$ | 0.3633 | - | 0.5945 | 0.7465 | 1.2758 | - |
| With FCN | 0.4124 | 2.3416 | 0.5720 | 0.6963 | 1.2524 | 5.0262 |

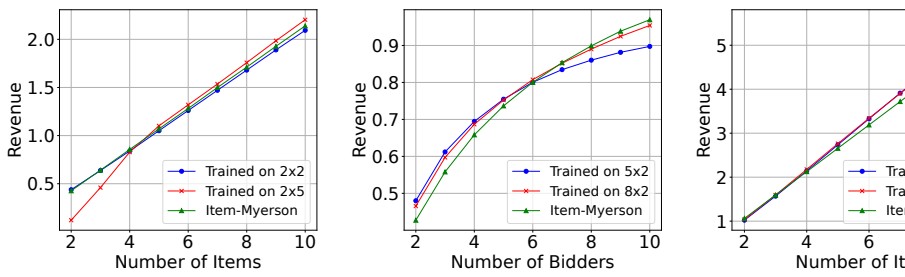

(a) Setting (A) with fixed $n = 2$.   (b) Setting (B) with fixed $m = 2$.   (c) Setting (C) with fixed $n = 3$.

Figure 2: Out-of-setting generalization results. We use the notation $n \times m$ to represent the number of bidders $n$ and the number of items $m$. We train AMenuNet and evaluate it on the same auction setting, excepts for the number of bidders or items. For detailed numerical results, please refer to Appendix C.

ability to approach near-optimal solutions in known settings (Setting (D)-(F)), even when those solutions are not AMAs themselves. This underscores the representativeness of the AMAs induced by AMenuNet. Remarkably, the comparison between AMenuNet and Lottery AMA in classic auctions showcases the effectiveness of training a neural network to compute AMA parameters. This suggests that AMenuNet captures the underlying mutual relations among the allocation menu, bidder weights, and boosts, resulting in a superior mechanism after training. Lastly, while CITransNet and RegretNet may achieve higher revenue in many cases, it is important to note that they are not DSIC. Such results indicate that AMenuNet's zero regret comes at the expense of revenue. Compared to RegretNet-based approaches, AMenuNet's distinctive strength lies in its inherent capacity to ensure DSIC by design.

**Ablation Study.** We present an ablation study that compares the full AMenuNet model with the following three ablated versions: AMenuNet with fixed $w = 1$, AMenuNet with fixed $\lambda = 0$, AMenuNet with $\mathcal{A}_{\mathrm{dtm}}$ (all the $(n + 1)^m$ deterministic allocations), and AMenuNet with the underlying architecture to be a 4-layers fully connected neural networks (FCN). We set the number of layers in FCN to be 4, each with 128 hidden nodes. The revenue results are presented in Table 2. For large values of $n$ and $m$, the experiment result of AMenuNet with $\mathcal{A}_{\mathrm{dtm}}$ can be intractable due to the large size of the allocation menu (59049 for $2 \times 10$ and 1048576 for $3 \times 10$). Such phenomenon indicates that we will face scalability issue if we consider all deterministic allocations. In contrast, the full model achieves the highest revenue in all cases except for $2 \times 10$(A), where it also performs comparably to the best result. Notice that, in Setting (C), $w = 1$ is coincidently the optimal solution due to the symmetry of the setting. The ablation results underscore the benefits of making the allocation menu, weights, and boost variables learnable, both for effectiveness and scalability. Moreover, even with significantly fewer learnable parameters, AMenuNetconsistently outperforms AMenuNetwith FCN, further highlighting the revenue advantages of a transformer-based architecture.

**Out-of-Setting Generalization.** The architecture of AMenuNet is designed to be independent of the number of bidders and items, allowing it to be applied to auctions with varying sizes. To evaluate such out-of-setting generalizability [Rahme et al., 2021a], we conduct experiments whose results are

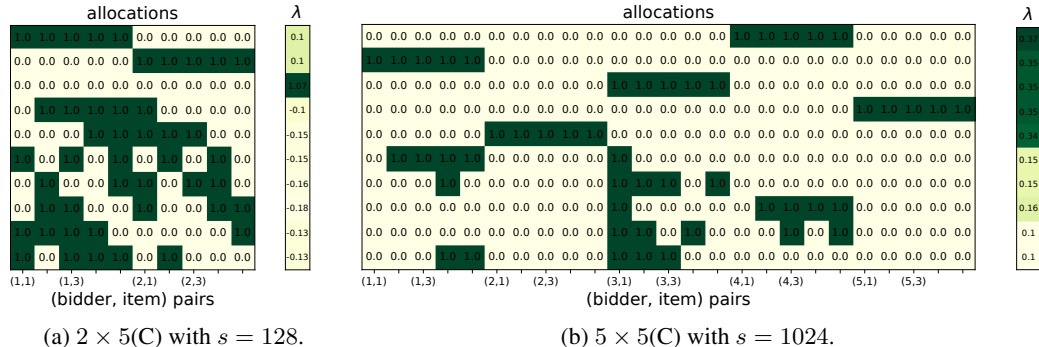

(a) $2 \times 5$(C) with $s = 128$.  (b) $5 \times 5$(C) with $s = 1024$.

Figure 3: The top-10 allocations (with respect to winning rate) and the corresponding boosts among $100,000$ test samples.

shown in Figure 2. The experimental results demonstrate that the generalized AMenuNet achieves revenue comparable to that of Item-Myerson, particularly in cases with varying items where AMenuNet often outperforms Item-Myerson. This highlights the strong out-of-setting generalizability of AMenuNet. Furthermore, The fact that AMenuNet can generalize to larger settings enhances its scalability.

**Winning Allocations.** We conducted experiments to analyze the winning allocations generated by AMenuNet. In order to assess the proportion of randomized allocations, we recorded the ratio of such allocations in the last row of Table 1a. Here, we define an allocation as randomized if it contains element in the range of $[0.01, 0.99]$. The results indicate that randomized allocations account for a significant proportion, especially in larger auction scales. For instance, in settings such as $2 \times 10$(A), $3 \times 10$(A), and $7 \times 2$(B), the proportion of randomized allocations exceeds $17\%$. Combining these findings with the results presented in Table 2, we observe that the introduction of randomized allocations leads to an improvement in the revenue generated by AMenuNet.

Additionally, we present the top-10 winning allocations based on their winning rates (i.e., allocations that are either $A^*$ or $A^*_{-k}$ for some $k \in [n]$) in Figure 3, specifically for the $2 \times 5$(C) and $5 \times 5$(C) settings. Notably, the top-10 winning allocations tend to be deterministic, suggesting that AMenuNet is capable of identifying useful deterministic allocations within the entire allocation space. Furthermore, in the $2 \times 5$(C) setting, we observed instances where the winning allocation was the empty allocation with a substantial boost. This can be seen as a reserve price, where allocating nothing becomes the optimal choice when all submitted bids are too small.

## 6  Conclusion and Future Work

In this paper, we introduce AMenuNet, a scalable neural network for the DSIC affine maximizer auction (AMA) design. AMenuNet constructs AMA parameters, including the allocation menu, bidder weights and boosts, from the public bidder and item representations. This construction ensures that the resulting mechanism is both dominant strategy compatible (DSIC) and individually rational (IR). By leveraging the neural network to compute the allocation menu, AMenuNet offers improved scalability compared to previous AMA approaches. The architecture of AMenuNet is permutation equivariant, allowing it to handle auctions of varying sizes and generalize to larger-scale auctions. Such out-of-setting generalizability further enhances the scalability of AMenuNet. Our various experiments demonstrate the effectiveness of AMenuNet, including its revenue, scalability, and out-of-setting generalizability.

As for future work, since we train AMenuNet using an offline learning approach, a potential direction is to explore online learning methods for training AMenuNet. Additionally, considering that the allocation menu size still needs to be predefined, it would be interesting to investigate the feasibility of making the allocation menu size learnable as well.

## Acknowledgments and Disclosure of Funding

This work is supported by the National Key R&D Program of China (2022ZD0114900). We thank all anonymous reviewers for their helpful feedback.

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

# A Proof of Theorem 4.1

**Theorem 4.1.** *The mechanism induced by AMenuNet satisfies both DSIC and IR.*

*Proof.* In AMenuNet, the computation of allocation menus $\mathcal{A}$, bidder weights $\boldsymbol{w}$ and allocation boosts $\boldsymbol{\lambda}$ are based on public bidder representation $X$ and item representation $Y$, without the chance to access the bids $B$. So we only have to prove that when $\mathcal{A}$, $\boldsymbol{w}$ and $\boldsymbol{\lambda}$ are fixed, the mechanism is both dominant strategy incentive compatibility (DSIC) and individually rational (IR).

**The proof of DSIC.** Denote by $u_k(\boldsymbol{v}_k, (\boldsymbol{v}_k, B_{-k}); X, Y)$ the utility bidder $k$ can get if she truthfully bids and $u_k(\boldsymbol{v}_k, (\boldsymbol{b}_k, B_{-k}); X, Y)$ the utility bidder $k$ can get if she bids $\boldsymbol{b}_k$. In fact, $u_k(\boldsymbol{v}_k, (\boldsymbol{v}_k, B_{-k}); X, Y)$ and $u_k(\boldsymbol{v}_k, (\boldsymbol{b}_k, B_{-k}); X, Y)$ can also be written as $u_k(\boldsymbol{v}_k, (\boldsymbol{v}_k, B_{-k}); \mathcal{A}, \boldsymbol{w}, \boldsymbol{\lambda})$ and $u_k(\boldsymbol{v}_k, (\boldsymbol{b}_k, B_{-k}); \mathcal{A}, \boldsymbol{w}, \boldsymbol{\lambda})$, respectively, since the parameters are determined by $X$ and $Y$.

Let $A^* := A^*((\boldsymbol{v}_k, B_{-k}); \mathcal{A}, \boldsymbol{w}, \boldsymbol{\lambda})$ and $A^*_{-k} := A^*_{-k}((\boldsymbol{v}_k, B_{-k}); \mathcal{A}, \boldsymbol{w}, \boldsymbol{\lambda})$ be the affine-welfare-maximizing allocation and the affine-welfare-maximizing allocation excluding bidder $k$, when bidder $k$ truthfully bids. Formally,

$$A^* = \arg\max_{A \in \mathcal{A}} \left( \sum_{i=1}^{n} w_i b_i(A) + \lambda(A) \right), A^*_{-k} = \arg\max_{A \in \mathcal{A}} \left( \sum_{i \neq k} w_i b_i(A) + \lambda(A) \right).$$

Similarly, we define $\widehat{A^*} := A^*((\boldsymbol{b}_k, B_{-k}); \mathcal{A}, \boldsymbol{w}, \boldsymbol{\lambda})$ and $\widehat{A^*_{-k}} := A^*_{-k}((\boldsymbol{b}_k, B_{-k}); \mathcal{A}, \boldsymbol{w}, \boldsymbol{\lambda})$ for bidder $k$ and her bids $\boldsymbol{b}_k$. Besides, let $p_k((\boldsymbol{v}_k, B_{-k}); \mathcal{A}, \boldsymbol{w}, \boldsymbol{\lambda})$ be bidder $k$'s payment defined by AMA. By definition,

$$p_k((v_k, B_{-k}); \mathcal{A}, \boldsymbol{w}, \boldsymbol{\lambda}) = \frac{1}{w_k} \left( \sum_{i \neq k} w_i b_i(A^*_{-k}) + \lambda(A^*_{-k}) \right) - \frac{1}{w_k} \left( \sum_{i \neq k} w_i b_i(A^*) + \lambda(A^*) \right).$$

First notice that $A^*_{-k}$ and $\widehat{A^*_{-k}}$ is the same since bidder $k$'s bid has no influence on others affine welfare. Hence,

$$\begin{aligned}
&u_k(\boldsymbol{v}_k, (\boldsymbol{v}_k, B_{-k}); X, Y) - u_k(\boldsymbol{v}_k, (\boldsymbol{b}_k, B_{-k}); X, Y) \\
=&u_k(\boldsymbol{v}_k, (\boldsymbol{v}_k, B_{-k}); \mathcal{A}, \boldsymbol{w}, \boldsymbol{\lambda}) - u_k(\boldsymbol{v}_k, (\boldsymbol{b}_k, B_{-k}); \mathcal{A}, \boldsymbol{w}, \boldsymbol{\lambda}) \\
=&v_k(A^*) - p_k((v_k, B_{-k}); \mathcal{A}, \boldsymbol{w}, \boldsymbol{\lambda}) - (v_k(\widehat{A^*}) - p_k((b_k, B_{-k}); \mathcal{A}, \boldsymbol{w}, \boldsymbol{\lambda})) \\
=&v_k(A^*) - \frac{1}{w_k} \left( \sum_{i \neq k} w_i b_i(A^*_{-k}) + \lambda(A^*_{-k}) \right) + \frac{1}{w_k} \left( \sum_{i \neq k} w_i b_i(A^*) + \lambda(A^*) \right) \\
&- v_k(\widehat{A^*}) + \frac{1}{w_k} \left( \sum_{i \neq k} w_i b_i(\widehat{A^*_{-k}}) + \lambda(\widehat{A^*_{-k}}) \right) - \frac{1}{w_k} \left( \sum_{i \neq k} w_i b_i(\widehat{A^*}) + \lambda(\widehat{A^*}) \right) \\
\overset{(1)}{=}&v_k(A^*) + \frac{1}{w_k} \left( \sum_{i \neq k} w_i b_i(A^*) + \lambda(A^*) \right) - v_k(\widehat{A^*}) - \frac{1}{w_k} \left( \sum_{i \neq k} w_i b_i(\widehat{A^*}) + \lambda(\widehat{A^*}) \right) \\
=&\frac{1}{w_k} \left( \sum_{i=1}^{n} w_i b_i(A^*) + \lambda(A^*) \right) - \frac{1}{w_k} \left( \sum_{i=1}^{n} w_i b_i(\widehat{A^*}) + \lambda(\widehat{A^*}) \right) \\
\overset{(2)}{\geq}&0,
\end{aligned}$$

where (1) holds by $A^*_{-k}$ and $\widehat{A^*_{-k}}$ is the same, and (2) holds by the definition of $A^*$.

**The proof of IR.** With the same notations above, we aim to prove that for any given representations $X, Y$ and others' bids $B_{-k}$, we have $u_k(\boldsymbol{v}_k, (\boldsymbol{v}_k, B_{-k}); X, Y) \geq 0$. We use the same notation

Table 3: The detailed parameters for all the settings (Setting (A)-(F)).

| Parameters | 2×2(A) | 2×5(A) | 2×10(A) | 3×2(A) | 3×5(A) | 3×10(A) |
|---|---|---|---|---|---|---|
| Menu Size $s$ | 32 | 64 | 64 | 256 | 256 | 256 |
| Temperature $\tau$ | 5 | 5 | 5 | 5 | 1 | 1 |
| Iterations | 3000 | 3000 | 3000 | 3000 | 8000 | 8000 |

| Parameters | 3×2(B) | 4×2(B) | 5×2(B) | 6×2(B) | 7×2(B) | 8×2(B) |
|---|---|---|---|---|---|---|
| Menu Size $s$ | 64 | 64 | 64 | 64 | 128 | 256 |
| Temperature $\tau$ | 3 | 3 | 3 | 3 | 1 | 1 |
| Iterations | 3000 | 3000 | 3000 | 8000 | 8000 | 8000 |

| Parameters | 2×5(C) | 3×10(C) | 5×5(C) | 3×1(D) | 1×2(E) | 1×2(F) |
|---|---|---|---|---|---|---|
| Menu Size $s$ | 128 | 512 | 1024 | 16 | 128 | 40 |
| Temperature $\tau$ | 10 | 10 | 50 | 10 | 10 | 10 |
| Iterations | 3000 | 2000 | 2000 | 2000 | 2000 | 2000 |

as in the proof of DSIC.

$$
\begin{aligned}
&u_k(\boldsymbol{v}_k, (\boldsymbol{v}_k, B_{-k}); X, Y) \\
=&u_k(\boldsymbol{v}_k, (\boldsymbol{v}_k, B_{-k}); \mathcal{A}, \boldsymbol{w}, \boldsymbol{\lambda}) \\
=&v_k(A^*) - p_k((v_k, B_{-k}); \mathcal{A}, \boldsymbol{w}, \boldsymbol{\lambda}) \\
=&v_k(A^*) - \frac{1}{w_k}\left(\sum_{i\neq k} w_i b_i(A^*_{-k}) + \lambda(A^*_{-k})\right) + \frac{1}{w_k}\left(\sum_{i\neq k} w_i b_i(A^*) + \lambda(A^*)\right) \\
=&\frac{1}{w_k}\left(\sum_{i=1}^n w_i b_i(A^*) + \lambda(A^*)\right) - \frac{1}{w_k}\left(\sum_{i\neq k} w_i b_i(A^*_{-k}) + \lambda(A^*_{-k})\right) \\
\overset{(1)}{\geq}&\frac{1}{w_k}\left(\sum_{i=1}^n w_i b_i(A^*) + \lambda(A^*)\right) - \frac{1}{w_k}\left(\sum_{i=1}^n w_i b_i(A^*_{-k}) + \lambda(A^*_{-k})\right) \\
\overset{(2)}{\geq}&0,
\end{aligned}
$$

where (1) holds by $w_k \geq 0$ and $v_k(\cdot) \geq 0$, and (2) holds by the definition of $A^*$. $\qquad\square$

## B  Further Implementation Details

**AMenuNet.** For all settings, the learning rate is $3 \times 10^{-4}$. We also apply a linear warm-up to the learning rate from $1 \times 10^{-8}$ to $3 \times 10^{-4}$ during the first 100 iterations. Totally in AMenuNet there are two softmax temperatures. Denote by $\tau_A$ the one in the softmax function for deciding social-welfare-maximizing allocation and $\tau$ the one used in Menu Layer to generate allocations. We set $\tau_A$ as 500 for all settings. We train the models for a maximum of 8000 iterations with $2^{15} = 32784$ new samples each iteration. For those we train more than 3000 iterations, we manually reduce the learning rate to $5 \times 10^{-5}$ after 3000 iterations. We test the model on a fixed test set with 100000 samples. The menu size and $\tau$ varies in different settings, and we present these numbers in 3. For classic auctions, including Setting (C)-(F), we embed each bidder and item ID into 16-dimension space. For encode layer, the output channel of the first $1 \times 1$ convolution in both the input layer and interaction modules are set to 64. The number of interaction modules is 3 or 5, and in each interaction module we adopt transformer with 4 heads and 64 hidden nodes. The boost layer is a two-layer fully-connected network with ReLU activation. To enable the out-of-setting generalization ability, we set both its channel and the hidden size as menu size.

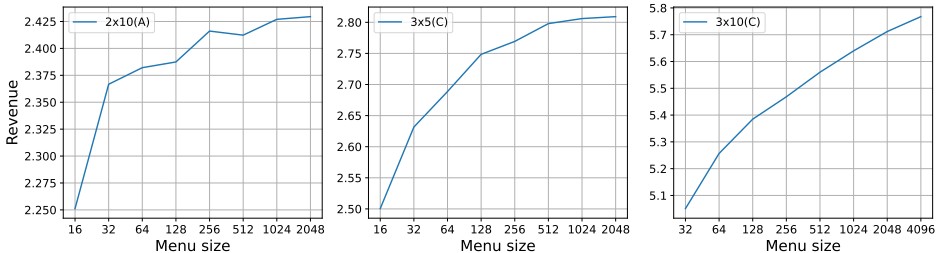

Figure 4: Revenue results of AMenuNet under different menu sizes.

Table 4: The experiment results of average revenue in out-of-setting generalization experiments.

| Method | 2×2(A) | 2×3(A) | 2×4(A) | 2×5(A) | 2×6(A) | 2×7(A) | 2×8(A) | 2×9(A) | 2×10(A) |
|---|---|---|---|---|---|---|---|---|---|
| Item-Myerson | 0.4265 | **0.6405** | **0.8559** | 1.0700 | 1.2861 | 1.4993 | 1.7111 | 1.9227 | 2.1398 |
| Trained on 2×2 | **0.4380** | 0.6371 | 0.8401 | 1.0503 | 1.2606 | 1.4706 | 1.6801 | 1.8900 | 2.0935 |
| Trained on 2×5 | 0.1212 | 0.4593 | 0.8286 | **1.1013** | **1.3193** | **1.5357** | **1.7579** | **1.9850** | **2.2034** |

| Method | 2×2(B) | 3×2(B) | 4×2(B) | 5×2(B) | 6×2(B) | 7×2(B) | 8×2(B) | 9×2(B) | 10×2(B) |
|---|---|---|---|---|---|---|---|---|---|
| Item-Myerson | 0.4274 | 0.5580 | 0.6584 | 0.7366 | 0.8004 | **0.8535** | **0.8990** | **0.9386** | **0.9696** |
| Trained on 5×2 | **0.4800** | **0.6120** | **0.6946** | **0.7547** | 0.8005 | 0.8346 | 0.8603 | 0.8815 | 0.8975 |
| Trained on 8×2 | 0.4656 | 0.5972 | 0.6868 | 0.7524 | **0.8076** | 0.8519 | 0.8902 | 0.9247 | 0.9542 |

| Method | 3×2(C) | 3×3(C) | 3×4(C) | 3×5(C) | 3×6(C) | 3×7(C) | 3×8(C) | 3×9(C) | 3×10(C) |
|---|---|---|---|---|---|---|---|---|---|
| Item-Myerson | **1.0628** | 1.5941 | 2.1256 | 2.6570 | 3.1883 | 3.7197 | 4.2512 | 4.7825 | 5.3140 |
| Trained on 3×10 | 1.0221 | 1.5686 | 2.1433 | 2.7365 | 3.3267 | **3.9116** | **4.4776** | **5.0440** | **5.5799** |
| Trained on 5×10 | 1.0366 | **1.5947** | **2.1766** | **2.7630** | **3.3398** | 3.9034 | 4.4598 | 5.0098 | 5.5498 |

**Item-Myerson.** There are no parameters in Myerson auction and VCG auction. We just implement the classic form of these two mechanisms. For Item-Myerson, we first compute the virtual value $\tilde{v}_i(v_i)$ according to $\tilde{v}_i(v_i) = v_i - \frac{1-F_i(v_i)}{f_i(v_i)}$, where $F_i$ and $f_i$ are the cdf and pdf function of $i$'s value distribution. Then allocate the item to the bidder with the highest virtual value if it is greater than 0. The payment by the winning bidder is equal to the minimal bid that she would have had to make in order to win.

**VCG.** VCG auction is a special case of AMA. It is equivalent to AMA by setting $\mathcal{A}$ as all deterministic allocations, $\boldsymbol{w}$ as 1 and $\boldsymbol{\lambda}$ as 0.

**Lottery AMA.** We use the implementation of Curry et al. [2023]. The hyperparameters and the results are mainly consistent with that reported in Curry et al. [2023]. We train all models for 10000 steps with $2^{15}$ newly generated samples. The learning rate is 0.001 and the softmax temperature is 100. We set menu size as $\{256, 4096, 2048, 16, 40, 40\}$ for the six experiments in Setting (C)-(F) respectively.

**CITransNet and RegretNet.** We follow the same hyperparameters in Duan et al. [2022] and Dütting et al. [2019].

## C  Further Experiments Results

**Impact of Menu Size.** In Figure 4, we present the experimental results of AMenuNet under various menu sizes. We conduct such experiment in setting $2 \times 10(A)$, $3 \times 5(C)$ and $3 \times 10(C)$. Generally, a larger menu, while entailing higher computational costs, can lead to increased revenue.

**Out-of-Setting Generalization.** In Table 4, we present the detailed numerical results of our out-of-setting generalization experiments.

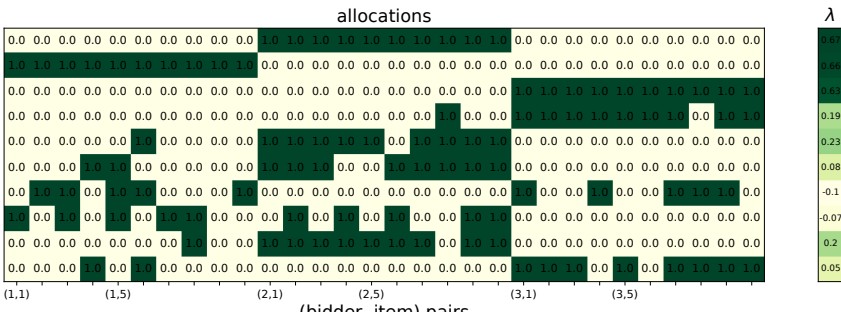

Figure 5: The top-10 allocations (with respect to winning rate) and the corresponding boosts among $100,000$ test samples in $3 \times 10(C)$, with menu size $s = 1024$.

**Winning Allocations.** We further present the top-10 winning allocations of $3 \times 10(C)$ in Figure 5. The experimental results maintain the same phenomenon: the top-10 winning allocations are all deterministic.

