# OpenReview forum: "A Scalable Neural Network for DSIC Affine Maximizer Auction Design"
_NeurIPS.cc/2023/Conference — NeurIPS 2023 spotlight_

### Official Review · Reviewer_1hHm · 2023-06-26

**Soundness:** 3 good
**Presentation:** 4 excellent
**Contribution:** 3 good
**Rating:** 7
**Confidence:** 5

**Summary:**

AMAs is a parametric family of auctions that generalizes VCG and that is exactly DSIC (unlike mechanisms found with alternative regret-based approaches) and IR. This paper proposes a deep variant of AMAs. In particular, AMAs parameters are learned as outputs of a permutation-equivariant attention-based network. Near-optimal mechanisms within AMA family can be found by optimizing this network. it can be applied to both contextual and classic auctions. The theory is provided to show that the learned auctions are indeed DSIC and IR. Extensive experiments and ablations support the method’s usefulness.

I will begin by saying that a deep variant of AMAs is a glaring gap in the literature, considering the popularity of deep regret-based approaches. Someone has to do it, and if done correctly, a paper like this absolutely should be published. Unfortunately, in my opinion, this particular version is not ready to be published yet.

**Strengths:**

- The paper is clear and well-written. It cites and discusses the relevant literature and places itself within the context of the literature.
- The goal of the paper, namely, a deep variant of AMAs, is worth pursuing. The paper is relevant to the conference.
- The experiments and ablations that are provided are extensive and look sound. I especially like the case study in Fig 3. I was surprised to see such diverse allocations, and the allocation acting as a reserve price is especially cool.
- The theory about the auctions being DSIC and IR is an essential part of the paper.


**Weaknesses:**

- My biggest worry is that experimental comparison is unfair. Authors claim in Table 1b that they “omit CITransNet since it is designed for contextual auctions” and proceed to compare their approach to RegretNet. RegretNet is based on fully-connected layers, whereas AMenuNet uses attention layers. Ivanov et al. (2022) show that attention layers improve revenue for the same regret levels in auctions with larger input sizes. So CITransNet (and, ideally, RegretFormer) should be included for comparison in auctions without context, as they are expected to achieve higher revenue than RegretNet and AMenuNet. For example, in the 3x10 auction, the paper reports 5.59 revenue of AMenuNet vs 5.54 revenue of RegretNet. Judging from these results, it might seem that AmenuNet is strictly better than RegretNet (same revenue but always DSIC). However, RegretFormer in its paper reports more than 6.1 revenue in 3x10 for the regret of <= 0.005 (as indicated in Table 1, “The regret of both CITransNet and RegretNet is less than 0.005”). By the way, this only takes 2 attention-based blocks, as opposed to 3-5 used by AMenuNet. If this is included in the comparison with AMenuNet, the conclusion would be that zero regret actually comes at a cost of revenue. This would not diminish the authors' contributions or the usefulness of their method, but would simply be a more sound experimental design.
- As Rahme et al. (2021) show, RegretNet is hyperparameter-sensitive. Choosing the wrong hyperparameters may decrease performance. The paper includes some settings that were not explored in the original RegretNet paper, like 3x1 and 5x5. However, in the appendix, it is only stated: “CITransNet and RegretNet. We follow the same hyperparameters in Duan et al. [2022] and Dütting et al. [2019].”. So for 3x1 and 5x5, it is not reported how the hyperparameters are selected, potentially resulting in unfair comparison. I would appreciate it if the authors could clarify this for me. As a suggestion, both Rahme et al. (2021) and Ivanov et al. (2022) propose alternative loss functions that are less sensitive to hyperparameters.
- A potential issue with AMenuNet is that it requires a hand-selected menu size. I wonder if larger settings would require larger menu sizes and if this introduces scaling issues. An ablation would be welcome where AMenuNet is given different menu sizes in some small and large settings and we examine revenue as a function of the menu size. E.g., we could see that the approach performs well in settings both large and small given the same small menu sizes, in which case there are no scaling issues.
- Compared to CITransNet or RegretFormer, there is nothing novel about the architecture, besides the softmax trick with annealed temperature to work with deterministic allocations. This is not a weakness per se, just a neutral fact.

**Questions:**

- How are hyperparameters selected for RegretNet in novel settings (that did not appear in its original paper)?
- Why do the authors think the learned menu is so diverse? If all allocations are trained on the same loss function, why is there no mode collapse (i.e., I could expect one “best” allocation to be repeated in all menu positions)?
- 176-177: “In non-contextual auctions, similar 177 to word embedding [Mikolov et al., 2013], we embed the unique ID of each bidder or item…”. This looks like positional encoding. Doesn’t it make learned mechanisms not permutation equivariant? The network could discriminate items and participants. Shouldn’t the same dummy ID be provided for all participants and items for true permutation equivariance?
- Why does AMenuNet outperform Lottery AMA, given that the parameters of AMAs are learned in both cases? We are used to attention outperforming everything else, but there is nothing to attend to in non-contextual auctions (as I understand, the input is always the same combination of IDs), so this is surprising to me.


**Limitations:**

For reasons mentioned in the weaknesses section, I think the method is a bit oversold.

---

> ### Author Rebuttal · Authors · 2023-08-08
>
> Thank you for your review! Your comments raise very good questions, and we will address them and the other concerns that you have raised.
>
> **Q1: About the experiments on RegretNet-based methods.**
>
> We understand the suggestion to include CITransNet and RegretFormer in the comparison for auctions without context, as they are expected to achieve higher revenue than RegretNet.
> However, it's important to note that our paper focus on comparing our methods against other DSIC approaches instead of the non-DSIC RegretNet-based algorithms.
> The comparison with RegretNet and CITransNet serves as supplementary insights rather than central facets of our study.
> We invite you to read the detailed discussion in Q1 of our general response for a more comprehensive discussion.
>
> > If this is included in the comparison with AMenuNet, the conclusion would be that zero regret actually comes at a cost of revenue.
>
> While incorporating CITransNet and RegretFormer in the comparison would indeed support the conclusion that "zero regret comes at a cost of revenue," our experimental results already demonstrate this trend:
>
> 1. In the contextual auction experiments, CITransNet outperformed AMenuNet in 9 of 10 instances.
> 2. In the classical auction experiments, RegretNet outperformed AMenuNet in 4 of 6 instances.
>
> **Q2: About the hyperparameters selection of RegretNet in novel settings**
>
> For 5x5(C), we set the hyperparameters of RegretNet as reported in the original paper for  '5x10 uniform' setting. Such choice was made due to the close relationship between these two cases. As indicated in Table 1(b), RegretNet with these hyperparameters performed well in 5x5(C), outperforming all other methods in terms of revenue.
>
> Similarly, for 3x1(D), we set the same configuration used for '1x2 uniform' in the original paper. Table 1(b) shows that RegretNet achieved revenue close to the optimal solution under these hyperparameters.
>
> **Q3: About the ablation study with respect to menu size.**
>
> We appreciate the suggestion for conducting an ablation study to analyze the impact of different menu sizes on AMenuNet's performance. We conduct experiments on AMenuNet using various menu sizes in two uniform (setting C) scenarios:
>
> | Menu Size    | 32     | 64     | 128    | 256    | 512    | 1024   | 2048    |
> | ------------ | ------ | ------ | ------ | ------ | ------ | ------ | --- |
> | 3x5 uniform  | 2.6336 | 2.7125 | 2.7470 | 2.7848 | 2.8051 | 2.8099 |  2.8110   |
> | 3x10 uniform | 5.1853 | 5.2348 | 5.4042 | 5.4932 | 5.5941 | 5.6714 |  5.7349   |
>
> We can see that AMenuNet's revenue performance improves as the menu size increases in both the 3x5 uniform and 3x10 uniform settings.
>
>
> **Q4: About why the learned menu is diverse.**
>
> The learned menu exhibits diversity for several reasons:
>
> 1. The menu includes allocations that are not only candidates for the winning allocation $A^*$ but also candidates for the allocation $A^*_{-k}$ that maximizes the affine social welfare for each bidder $k \in [n]$. This requirement for non-zero revenue leads to at least two different allocations within the menu, resulting in no unique allocation.
> 2. Although all allocations are trained using the same loss function, gradients for each allocation within a training batch differ. The use of the softmax function to approximate the winning allocation $\hat{A}^*$ during training means that only the best allocation (i.e., the one with the highest score) receives a significant gradient during backpropagation.
> 3. The diversity is further enhanced by sampling valuation profiles from a prior distribution during training. Different valuation profiles within the training batch can lead to various best allocations.
>
> **Q5: About the embeddings of the IDs**
>
> To ensure the mechanism can handle diverse and heterogeneous participants and items effectively, the use of unique ID embeddings is essential.
> For instance, in Myerson auctions [Myerson, 1981], the auctioneer relies on prior knowledge of each bidder's valuation profile, making it essential to correctly identify each bidder for accurate prior distribution estimation.
>
> Using the same dummy ID for all participants and items would hinder the mechanism's ability to differentiate between different bidders and items, especially in cases of heterogeneity. Consequently, the induced auction mechanism would be limited to symmetric settings, as seen in Rahme et al. 2021.
>
>
> **Q6: About the comparison between AMenuNet and Lottery AMA.**
>
> We discussed the potential reasons for AMenuNet's superior performance compared to Lottery AMA in Q2 of our general response.
> In short, AMenuNet's neural network effectively captures the correlation of AMA parameters, and its over-parameterization provides advantages for optimization, leading to improved results.

---

> > ### Comment · Reviewer_1hHm · 2023-08-10
> > **Rebuttal response**
> >
> > Thank you for the detailed answers to my questions!
> >
> > 1. I understand your position and let me state again that I do think the paper makes a valuable contribution.
> >
> > That said, comparing with relevant baselines is essential, and I guess we disagree on what are the relevant baselines in this case. I would not mind as much if it only was the case that these baselines are likely to outperform the method in question. But they are using similar attention-based architectures. And the paper does compare with RegretNet (as it should) which uses a "weaker" architecture.
> >
> > Regarding RegretNet outperforming AMenuNet 4 out of 6 times, the absolute difference is small, and there are 2 out of 6 times when the reverse is true. I am not even sure the result is statistically significant.
> >
> > At the very least, I advise including metrics reported in the respective papers of EquivariantNet, RegretNet, and CITransNet, for the settings that intersect with your paper. Also, I advise including an ablation where attention layers in AMenuNet are replaced with fully-connected layers. If this performed worse (although either outcome would be interesting), it would also show that attention > fully-connected. In conjunction, these two changes (and an explicit discussion), would be a sufficient alternative.
> >
> > 2. This is reasonable, thank you for clarifications! I suggest adding this to the appendix.
> >
> > 3. This is very interesting, thanks! I also wonder about a smaller setting like 2x2 but I leave this consideration with the authors.
> >
> > 4. and 6. I see, thank you for the clarification!
> >
> > 5. I agree with every word in your response. I understand why distinguishing agents is essential in asymmetric settings. But I still think this makes the architecture non-permutation-equivariant, and this has not been addressed in the rebuttal.
> >
> > I will elaborate. Consider a setting with two participants. The first has an id "1" and its valuation (for all items) is sampled from U[0, 1]. The second has an id "2" and its valuation is sampled from U[0, 2]. Yes, not providing an id would not learn an optimal auction, but it would be symmetric, and the architecture (let's say, an attention layer) -- permutation-equivariant. If we encode positions "1" and "2" before the attention layers, the optimal auction is in the space of solutions so it can be learned, but the architecture is no longer permutation-equivariant. Do we suddenly get the best of both worlds when we call "1" and "2" not positions but ids? No, we learn a mechanism that is not symmetric. We effectively encoded positions. It does not matter that technically the attention layers are permutation-equivariant. In the prior literature, permutation-equivariant architecture == symmetric mechanisms, and non-permutation-equivariant architecture == asymmetric mechanisms. So it is confusing to me that the method is claimed to be permutation-equivariant while learning asymmetric mechanisms.
> >
> > On top of that -- what's so great about permutation equivariance in this case? The paper cites Quin et al. [2022], but they "consider the popular additive valuation and symmetric valuation setting". The EquivariantNet also only experiments with symmetric settings. And the property to generalize (lines 340-341) is not due to equivariance -- NLP routinely applies positional encodings (order matters) that work for sentences of varying lengths. The paper treats attention-based architecture as equivariant regardless of the positional (or id) encoding. So answering the question of what's so great about equivariance is actually not easy -- is it the equivariance or the attention layers that are great? It could be answered with ablations with fully-connected layers (like RegretNet) and deepset layers (like EquivariantNet) since both don't use attention and only the latter is equivariant. But this might be too deep of a rabbit hole to prove something about the property that AMenuNet does not even possess (as argued above).
> >
> > To be clear, this is a nitpick, but the paper seems to make a big deal out of equivariance, and I do not think it should.
> >
> >
> >
> > To conclude, I still believe the paper oversells and the soundness should be improved. The rebuttal did not change my opinion. I leave my score unchanged and I leave it to ACs to decide how important these concerns are. But feel free to let me know if I missed/misinterpreted something.

---

> > > ### Author Response · Authors · 2023-08-13
> > > **Thank you for your comments!**
> > >
> > >
> > > Thank you for your comments! We will clarify the following concerns you raised.
> > >
> > > **Q7: The addtional experiments.**
> > >
> > > >  But they are using similar attention-based architectures. And the paper does compare with RegretNet (as it should) which uses a "weaker" architecture.
> > >
> > > We agree that CITransNet and AMenuNet share similar attention-based architectures.
> > > This is the reason why we conducted experiments to compare CITransNet and AMenuNet in Table 1(a).
> > > As for similar attention-based architectures in classical auctions, we addtionally conduct supplementary experiments in CITransNet, where we treat the IDs as discrete contexts.
> > > We report CITransNet's revenue when the regret is slightly under 5e-4 in the following table, where the other results are taken from our paper:
> > >
> > > | Settings   | DSIC? | 2x5(C)     | 3x10(C)    | 5x5(C)     | 3x1(D)     | 1x2(E)     | 1x2(F)     |
> > > | ---------- | ----- | ---------- | ---------- | ---------- | ---------- | ---------- | ---------- |
> > > | Optimal    | Yes   | -          | -          | -          | 2.7490     | **9.7810** | **0.1706**     |
> > > | CITransNet | No    | **2.3788** | **5.9191** | **3.4759** | **2.7541** | 9.7551     | 0.1691 |
> > > | RegretNet  | No    | 2.3390     | 5.5410     | 3.4260     | 2.7264     | 9.7340     | 0.1732     |
> > > | AMenuNet   | Yes   | 2.2768     | 5.5896     | 3.3916     | 2.7382     | 9.6219     | 0.1701     |
> > >
> > > We observe that the attention-based CITransNet consistently outperforms both RegretNet and AMenuNet across all scenarios with unknown optimal solutions. Additionally, CITransNet approximates the optimal solution well in the known cases. These outcomes underscore the revenue advantages of the attention module in CITransNet when contrasted with the fully connected neural network-based RegretNet. Furthermore, the results reinforce the conclusion that AMenuNet's zero regret comes at the expense of revenue.
> > >
> > > However, it's important to emphasize that the advantages of AMenuNet compared to CITransNet and RegretNet are not primarily focused on revenue. Instead, AMenuNet's distinctive strength lies in its inherent capacity to ensure Dominant Strategy Incentive Compatibility (DSIC) by design.
> > >
> > > > Regarding RegretNet outperforming AMenuNet 4 out of 6 times, the absolute difference is small, and there are 2 out of 6 times when the reverse is true. I am not even sure the result is statistically significant.
> > >
> > > As we described at the beginning of experiment section, all of our presented results are the average of the results of experiments in five different seeds.
> > >
> > >
> > > > At the very least, I advise including metrics reported in the respective papers of EquivariantNet, RegretNet, and CITransNet, for the settings that intersect with your paper.
> > >
> > > In our paper, we have already included the revenue reported in RegretNet and CITransNet for the intersect settings.
> > >
> > > > Also, I advise including an ablation where attention layers in AMenuNet are replaced with fully-connected layers.
> > >
> > >
> > > We agree that conducting an ablation study to investigate the revenue benefits of attention layers compared to fully-connected layers is indeed interesting. However, we believe this experiment is better suited for inclusion in the Appendix rather than the main body.
> > >
> > > This is because our primary motivation behind employing attention-based layers extends beyond merely enhancing revenue performance. Our goal is to establish permutation-equivariance and the ability to generalize to auctions with varying scales, ultimately enhancing scalability. These specific attributes are not satisfied by fully-connected layers.
> > >
> > > For the ablation experiment, we compare AMenuNet with AMenuNet-FCN, wherein we replace the transformer-based interaction modules with a multi-layer fully connected neural network. We set the number of layers in AMenuNet-FCN to $4$, each with $128$ hidden nodes.
> > >
> > > | Settings   | DSIC? | 2x5(C)     | 3x10(C)    | 5x5(C)     |
> > > | ---------- | ----- | ---------- | ---------- | ---------- |
> > > | AMenuNet   | Yes   | **2.2768**     | **5.5896**     | **3.3916**     |
> > > |   AMenuNet-FCN     |   Yes |  2.1333     |      5.0161      |      3.3657      |
> > >
> > > We can observe that AMenuNet consistently outperforms AMenuNet-FCN across the listed scenarios. Additionally, the parameter count of AMenuNet is lower than AMenuNet-FCN's, further underscoring the revenue advantages of a transformer-based architecture.
> > >
> > > We will consider adding the ablation experiments in the Appendix.

---

> > > ### Author Response · Authors · 2023-08-13
> > > **About permutation-equivariance.**
> > >
> > > **Q8: About permutation-equivariance.**
> > >
> > > There is a disagreement over the exact meaning of permutation-equivariance. The key is **whether the permutation operator should rearrange bidder and item IDs or contexts**. We must highlight that our paper does not treat the IDs as fixed positional encoding. Instead, **as described in Definition 4.2, we will also permute the IDs when we permute the bids.**
> > >
> > > **--Q8.1: The definition of permutation-equivariance.**
> > >
> > > >  The paper treats attention-based architecture as equivariant regardless of the positional (or id) encoding.
> > >
> > > Your understanding of permutation-equivariance assumes that the permutation operator rearranges bids alone. However, this interpretation mainly fits symmetric auction situations. Here, specific bidder or item IDs aren't crucial, so there's no need to account for their permutation.
> > >
> > > On the other hand, our concept of permutation-equivariance, formally laid out in Definition 4.2, has a broader scope. It encompasses scenarios where public IDs or contexts of bidders and items have significant impacts. Therefore, the permutation process should also encompass shuffling these IDs. This definition aligns with Duan et al. [2022] and Qin et al. [2022], which we'll delve into later.
> > >
> > > **--Q8.2: About the example of 2 bidders and 1 item, with $v_1 \sim U[0, 1]$ and $v_2 \sim U[0, 2]$.**
> > >
> > > To clarify, in scenarios considering IDs, AMenuNet treats the ID "1" as "2" in its inputs.
> > > If we swap the bidders, the order of their IDs *will also switch*, because IDs hold significance as public information.
> > > Therefore, the outcome of allocation and payment will maintain the same permutation as the input bids and IDs.
> > >
> > > **--Q8.3: Permutation-equivariance in previous literatures.**
> > >
> > > > In the prior literature, permutation-equivariant architecture == symmetric mechanisms, and non-permutation-equivariant architecture == asymmetric mechanisms.
> > >
> > > We disagree with the given statement. Symmetric mechanisms can be considered a specific case of permutation-equivariance. Permutation equivariance can also apply to asymmetric mechanisms as long as the permutation operates on bidder IDs (or contexts).
> > >
> > > In the existing literature, researchers have introduced the concept of permutation equivariance to cover asymmetric auctions by permuting IDs (or contexts). For instance, Duan et al. [2022] define permutation-equivariance in Remark 3.1 as follows:
> > >
> > > > [Remark 3.1 of Duan et al. [2022]] We say an auction mechanism $(g^w, p^w)$ is permutation-equivariant if for any two permutation matrices $\Pi_{n}\in \{0,1\}^{n\times n}$ and $\Pi_{m}\in \{0,1\}^{m\times m}$, and any input (including bids $b \in \mathbb{R}^{n\times m}$, bidder-contexts $x \in \mathbb{R}^{n\times d_x}$ and item-contexts $y \in \mathbb{R}^{m\times d_y}$), we have $g^w(\Pi_{n}b\Pi_{m}, \Pi_n x, \Pi_m^T y)=\Pi_{n}g^w(b,x,y)\Pi_{m}$ and $p^w(\Pi_{n}b\Pi_{m}, \Pi_n x, \Pi_m^T y)=\Pi_{n}p^w(b,x,y)$
> > >
> > > Furthermore, Qin et al. [2022] also incorporate the concept of permutation applied to the bidder and item IDs or contexts, despite their focus on additive and symmetric valuations. This is evident in their definitions of bidder orbit averaging $\mathcal{Q}_1$ and item averaging $\mathcal{Q}_2$:
> > >
> > > > [Qin et al. [2022]] The bidder averaging $\mathcal{Q}_1$ and the item averaging $\mathcal{Q}_2$ acting on the allocation rule $g$ and the payment rule $p$, respectively, are as below,
> > > > $$\mathcal{Q}_1{g}(v,x,y)=\frac{1}{n!}\sum _{\sigma _n\in S_n}\sigma _n^{-1}g(\sigma_n v,\sigma_n x,y), \mathcal{Q} _1{p}(v,x,y)=\frac{1}{n!}\sum _{\sigma _n\in S_n}\sigma _n^{-1}p(\sigma _n v,\sigma _n x,y),$$
> > > > $$\mathcal{Q}_2{g}(v,x,y)=\frac{1}{m!}\sum _{\sigma_m\in S_m}g(v \sigma_m,x,y\sigma_m)\sigma_m^{-1}, \mathcal{Q}_2{p}(v,x,y)=\frac{1}{m!}\sum _{\sigma_m\in S_m}p(v\sigma_m,x,y\sigma_m).$$
> > >
> > > Such definition clearly shows the application of permutation to bidder contexts $x$ and item contexts $y$.
> > >
> > >
> > > In summary, it is important to recognize that permutation-equivariance goes beyond symmetric auctions. It can be extended to include asymmetric auctions by employing permutations on bidder IDs or contexts.
> > >
> > >
> > > **References**
> > >
> > > [1] Zhijian Duan, Jingwu Tang, Yutong Yin, Zhe Feng, Xiang Yan, Manzil Zaheer, and Xiaotie Deng. A context-integrated transformer-based neural network for auction design. ICML 2022.
> > >
> > > [2] Tian Qin, Fengxiang He, Dingfeng Shi, Wenbing Huang, and Dacheng Tao. Benefits of permutation equivariance in auction mechanisms. NeurIPS 2022.

---

> > > > ### Comment · Reviewer_1hHm · 2023-08-13
> > > >
> > > > Thank you for the clarifications! I do appreciate the authors' effort to address my concerns and, especially, to present new results, in such a short time period. I feel like my concerns are addressed, please include new results in the main paper or the appendix. I will raise my score.

---

> > > > > ### Author Response · Authors · 2023-08-14
> > > > > **Appreciation for Your Feedback!**
> > > > >
> > > > > Thank you for your encouraging feedback! We will incorporate the new experimental results into the revision. Thank you once again!

---

### Official Review · Reviewer_MRew · 2023-06-30

**Soundness:** 4 excellent
**Presentation:** 4 excellent
**Contribution:** 4 excellent
**Rating:** 8
**Confidence:** 4

**Summary:**

This paper introduces AmenuNet, a scalable NN for the AMA design, which ensures DSIC and IR. And the experiments demonstrate the effectiveness of AMenuNet, including its revenue, scalability, and out-of-setting generalizability.

**Strengths:**

Originality:
1.Proposes a new automated auction design method: The paper proposes a new automated auction design method that uses a neural network to construct AMA parameters, thereby improving scalability and revenue performance.
2.Combines deep learning and game theory: The paper combines deep learning and game theory to propose a new method for solving the DSIC problem in multi-item auctions, which is a novel approach.

Quality:
1.Experimental results demonstrate the effectiveness of the proposed method: The paper demonstrates the effectiveness of the proposed method in extensive experiments, including performance in different environments and scalability in large auctions.
2.Theoretical proof ensures the DSIC property of the auction: The paper ensures the DSIC property of the auction through theoretical proof, thereby improving the quality of the auction.

Clarity:
1.Clear paper structure: The paper has a clear structure, strong logic, and is easy to understand.
2.Detailed experimental section: The experimental section of the paper is detailed, including experimental settings, results, and analysis, making it easy for readers to understand and reproduce.

Importance:
1.Contribution to the field of automated auction design: The proposed method in the paper makes an important contribution to the field of automated auction design, helping people design high-revenue auction mechanisms.
2.Inspirational significance for the combination of deep learning and game theory: The paper explores the combination of deep learning and game theory, providing inspiration for related research in the field.

**Weaknesses:**

1.Lack of comparison with existing methods: The paper does not compare the proposed method with some existing industrial methods, such as DNA, NMA, which makes it difficult to evaluate the novelty and effectiveness of the proposed method in industry.
2.Limited applicability: The proposed method is only verified on small-scale data sets, lacking demonstrations on large-scale industrial scenarios.
3.Expression problem: some typos need to be optimized.

**Questions:**

1.Lack of modeling of full contextual externalities.
2.Lack of comparison with WVCG, NMA.

**Limitations:**

No.

---

> ### Author Rebuttal · Authors · 2023-08-08
>
> Thank you for your constructive comments! We appreciate your positive feedback and we will address the questions you listed.
>
> **Q1: About the comparison with industrial methods.**
>
> > The paper does not compare the proposed method with some existing industrial methods, such as DNA, NMA, which makes it difficult to evaluate the novelty and effectiveness of the proposed method in industry.
>
> > Lack of comparison with WVCG.
>
> In case of misunderstanding, we refer DNA as the method proposed in the paper "Neural auction: End-to-end learning of auction mechanisms for e-commerce advertising", NMA as the one proposed in the paper "NMA: Neural Multi-slot Auctions with Externalities for Online Advertising" and WVCG as the one presented in the paper "Truthful learning mechanisms for multi-slot sponsored search auctions with externalities."
>
> It's worth noting that all of these methods focus on a different type of auction problem, namely, advertising auction.
> Advertising auctions are single-parameter auctions, which involve bidders placing a single parameter as their valuation.
> The DSIC characterization of such auctions is based on Myerson Lemma [Myerson, 1981].
>
> In contrast, our AMenuNet is specifically designed for more complex multi-parameter auctions, where bidders bid multiple parameters as their valuations.
> Such auctions introduce greater complexity due to the lack of DSIC characterizations, making them distinct from single-parameter auction settings.
> Therefore, while DNA, NMA and WVCG are relevant for advertising auctions, the nature of the auction setting addressed by AMenuNet is inherently different.
>
> **Q2: About the scalability.**
>
> > The proposed method is only verified on small-scale data sets, lacking demonstrations on large-scale industrial scenarios.
>
> As highlighted earlier in Q1, our primary focus is on multi-parameter auctions, a more intricate setting than industrial advertising auctions. The complexity inherent to multi-parameter auctions poses challenges for scaling to large industrial scenarios.
> Notably, the auction scales we've considered in this paper already exceed those of prior AMA-based methods, such as Sandholm and Likhodedov [2015] and Curry et al. [2022], and are comparable to previous RegretNet-based approaches.
>
> **References**
> [1] Tuomas Sandholm and Anton Likhodedov. Automated design of revenue-maximizing combinatorial auctions. Operations Research, 63(5):1000–1025, 2015
> [2] Michael Curry, Tuomas Sandholm, and John Dickerson. Differentiable economics for randomized affine maximizer auctions. IJCAI 2023.

---

### Official Review · Reviewer_Dfp6 · 2023-07-06

**Soundness:** 4 excellent
**Presentation:** 4 excellent
**Contribution:** 2 fair
**Rating:** 5
**Confidence:** 5

**Summary:**

This paper proposes a new architecture for learning auctions that are DSIC (but not necessarily revenue optimal) auctions. The authors' main contribution lies in a transformer-based permutation equivariant architecture designed to calculate the allocations, weights, and boosts variables utilized by AMA-based approaches.

**Strengths:**

The primary strength of this novel architecture resides in its capacity to effectively handle the following aspects:

1. **Contextual and non-contextual auctions:** The architecture demonstrates the ability to accommodate both contextual and non-contextual auction settings, allowing for a broader range of applications.
2. **Equivariance for varying auction sizes:** The architecture exhibits permutation equivariance, enabling it to handle auctions of different sizes without requiring significant modifications.
3. **Better scalability:** In comparison to existing approaches, the proposed architecture exhibits improved scalability, providing a more efficient and scalable solution for auction learning tasks.

**Weaknesses:**

This paper shares similarities with the work of Curry et al. [2022] on Differential Economics for Randomized AMA auctions. However, a key distinction is that the allocation menus in this paper are parameterized by a neural network, in contrast to directly being parameterized through autograd variables. Additionally, the proposal of contextual auction design through the transformer architecture has been previously suggested by Dual et al. [2022]. Consequently, the contributions of this paper may not be considered highly novel. Nevertheless, the authors demonstrate improved revenue performance over existing approaches, as evidenced in Table 2.

**Questions:**

In the context of classical auctions, is there any intuition for why optimizing the neural network that outputs an allocation works better than directly optimizing the allocation variables?

---

> ### Author Rebuttal · Authors · 2023-08-08
>
> Thank you for your constructive review! We will address the questions you have listed.
>
> **Q1: About the contribution**
>
> > This paper shares similarities with the work of Curry et al. [2022] on Differential Economics for Randomized AMA auctions. However, a key distinction is that the allocation menus in this paper are parameterized by a neural network, in contrast to directly being parameterized through autograd variables. Additionally, the proposal of contextual auction design through the transformer architecture has been previously suggested by Dual et al. [2022]. Consequently, the contributions of this paper may not be considered highly novel.
>
> In contrast to the work of Curry et al. [2022] (Lottery AMA), our contribution lies in introducing a novel deep neural network-based AMA method. The underlying neural network architecture empowers our method to handle classical and contextual settings, offering flexibility and adaptability.
> Furthermore, our method enjoys several advantages beyond revenue maximization. Its inherent permutation equivariance ensures consistent outcomes regardless of the order of bidders or items, enhancing its practicality. Additionally, the scalability of our approach is noteworthy, as the neural network can effectively process input data of various sizes, making it applicable to a wide range of auction scenarios.
>
>
> Compared to the work of Duan et al. [2022], our AMA-based method can ensure Dominant-Strategy Incentive Compatibility (DSIC) by construction.
> This property enhances the reliability and trustworthiness of the proposed mechanism, providing bidders with strong incentives to reveal their true valuations.
>
>
> **Q2: About the benefits of AMenuNet with respect to lottery AMA.**
>
> > In the context of classical auctions, is there any intuition for why optimizing the neural network that outputs an allocation works better than directly optimizing the allocation variables?
>
> We have extensively discussed the reasons for AMenuNet's superior performance over Lottery AMA in our general response (Q2). In summary, the key factors contributing to AMenuNet's advantage are:
> 1. AMenuNet adds more inductive bias by capturing the correlation of AMA parameters through the neural network.
> 2. AMenuNet's over-parameterization from the neural network offers a better optimization landscape.

---

> > ### Comment · Reviewer_Dfp6 · 2023-08-14
> > **Response to the rebutatal**
> >
> > 1. Can you please elaborate on what's novel in the neural network-based AMA method - is this a simple change in the output layer to output menus or something different?
> > 2. In the case of non-contextual auctions, I understand that the inputs are fixed for a given setting. In this case, are you claiming/ observing that optimizing a parameterized function to generate a set of outputs is easier than directly optimizing the outputs itself?
> > 3. Regarding permutation equivariance, how do you achieve permutation symmetry concerning the bids once you have the allocations and boosts? Can you use similar techniques with Lottery AMA parameters as well?

---

> > > ### Author Response · Authors · 2023-08-15
> > > **Thank you for your further comments!**
> > >
> > >
> > > Thank you for your further comments! We will address the questions you have listed.
> > >
> > > **C1**
> > > > 1. Can you please elaborate on what's novel in the neural network-based AMA method - is this a simple change in the output layer to output menus or something different?
> > >
> > >
> > > First, we need to highlight that, to the best of our knowledge, we are the first to incorporate a neural network into AMAs. In contrast, previous literature directly provides and optimizes these AMA parameters without the usage of neural networks. Consequently, their methods are limited to classical auctions.
> > >
> > > In our paper, we are not "simply changing the output layer to output menus". Instead, we are the first to use the neural network to output all the AMA parameters (the allocation menu, weights, and boosts). As a result, our method can handle both classical and contextual auctions.
> > >
> > > Another novelty is the construction of our attention-based neural network, AMenuNet. The weights of AMenuNet are not affected by the number of bidders and items. Therefore, AMenuNet can generalize to auction settings with a different number of bidders and items, enhancing its scalability.
> > >
> > >
> > > **C2**
> > > > 2. In the case of non-contextual auctions, I understand that the inputs are fixed for a given setting. In this case, are you claiming/ observing that optimizing a parameterized function to generate a set of outputs is easier than directly optimizing the outputs itself?
> > >
> > > In non-contextual auctions, we observe in experimental results that optimizing the neural network is easier than optimizing the AMA parameters. We have discussed the possible reasons in Q2 of our general response.
> > >
> > > In short, AMenuNet provides more inductive bias by capturing the correlation between different AMA parameters through the neural network. Moreover, AMenuNet's over-parameterization potentially offers a better optimization landscape.
> > >
> > >
> > > **C3**
> > > > 3. Regarding permutation equivariance, how do you achieve permutation symmetry concerning the bids once you have the allocations and boosts? Can you use similar techniques with Lottery AMA parameters as well?
> > >
> > > As shown in Definition 4.2, if we permute the bidders and items, **both the bids and IDs (or contexts) of all bidders and items will be permuted**. Since **the IDs (or contexts) are the inputs of AMenuNet**, once we permute them, the output of AMenuNet will also be permuted in the same way due to its permutation-equivariance architecture. Our definition of permutation-equivariance is also used in Duan et al. 2022 and Qin et al. 2022, and we have discussed it in detail in Q8 of our response to Reviewer 1hHm.
> > >
> > > For the Lottery AMA, it directly provides the allocations and boosts regardless of the permutation of IDs.
> > > Therefore, the same technique cannot be applied to the Lottery AMA.
> > >
> > > **References**
> > >
> > > [1] Zhijian Duan, Jingwu Tang, Yutong Yin, Zhe Feng, Xiang Yan, Manzil Zaheer, and Xiaotie Deng. A context-integrated transformer-based neural network for auction design. ICML 2022.
> > >
> > > [2] Tian Qin, Fengxiang He, Dingfeng Shi, Wenbing Huang, and Dacheng Tao. Benefits of permutation equivariance in auction mechanisms. NeurIPS 2022.

---

### Official Review · Reviewer_Zst9 · 2023-07-07

**Soundness:** 4 excellent
**Presentation:** 3 good
**Contribution:** 4 excellent
**Rating:** 8
**Confidence:** 5

**Summary:**

Revenue maximizing strategyproof auction design with multidimensional types has proven to be extremely challenging. The lack of theoretical progress even in simple problem instances has motivated the use of machine-learning-based techniques to approximately learn high-performing auctions. One approach, typified by “RegretNet” and its followup works, involves the use of neural networks as function approximators to directly represent auction mechanisms — the training process optimizes revenue and strategyproofness. However, although the mechanisms learned by these neural network based approaches are qualitatively good and likely near to the true optimal mechanism, the enforcement of the strategyproofness constraints is not perfect — a significant limitation. Another line of work searches within some restricted class of mechanisms, all of which are guaranteed to be strategyproof, for one that performs well.

The current work builds on this latter approach. Like some previous work, they focus on the class of affine maximizer auctions, which are guaranteed to be strategyproof. However, previous work directly optimized the parameters and possible outcomes of the auction. They instead treat the mechanism itself as the outcome of a neural network. Crucially, this neural network does not see the bids as input — at most, it sees some informative “context” features — so strategyproofness is preserved. Also, even for auctions without context, training using this architecture seems to perform better than optimizing the parameters directly. The neural network uses a transformer architecture, which comes with some advantages — it allows for permutation-equivariance, a useful property which is satisfied by optimal auctions when the bidders are anonymous.

In experiments, the authors find that their method performs very well -- getting higher revenue than previous strategyproof approaches and even performing comparably to the unconstrained neural network approaches in some cases.


**Strengths:**

The experiments are done quite well and follow standard methodology, the baselines are well-chosen, the method is explained clearly, the results show a clear improvement on existing work, and the technique opens up capabilities in new settings.

**Weaknesses:**

There have since been many improvements on RegretNet in addition to CITransNet, with and without contexts. It could be interesting to compare to some of these as well.

The section describing the auction architecture is extremely dense. It's possible this had to be compressed a lot for the NeurIPS page limit, but I found it hard to follow. I ended up looking at the diagram+code and found this much easier to understand than the written part.

**Questions:**

Where do you get the reported revenues for the other methods? Are these experiments you ran yourself or taken from another paper? If reproduced yourself, how do they compare to reported results in other papers?

Your table 1 has some methods in bold even when RegretNet/CITransNet outperform them. I admit it does say in the caption that bold means “best among DSIC methods”, and the DSIC methods are separated by bars, but when reading the paper, the first thing I did was immediately look at that table, so I still found it confusing. I think it would be good to find a way to make this even more obvious.

**Limitations:**

The authors adequately discuss the limitations of their approach (the main one being that it deals with a restricted class of auctions, unlike the RegretNet approaches).

---

> ### Author Rebuttal · Authors · 2023-08-08
>
> Thank you for your positive feedback! We value your affirmation and we will address the questions and concerns you have listed.
>
> **Q1: About the comparison of RegretNet-based approaches.**
>
> > There have since been many improvements on RegretNet in addition to CITransNet, with and without contexts. It could be interesting to compare to some of these as well.
>
> While introducing more RegretNet-based methods into our experiments could be interesting, our primary focus is comparing our method with other DSIC (Dominant-Strategy Incentive-Compatible) approaches.
> We use the comparison with RegretNet and CITransNet to give extra information, not to make them the main focus.
> We suggest reading the detailed discussion in Q1 of our general response.
>
> **Q2: About the results of baselines.**
>
> > Where do you get the reported revenues for the other methods? Are these experiments you ran yourself or taken from another paper? If reproduced yourself, how do they compare to reported results in other papers?
>
> For the reported revenues of other methods, we used their previously reported results for settings that appeared in other papers. For settings not found in prior work, we conducted our own experiments. You can find detailed descriptions of hyperparameter settings in Appendix B.
>
> **Q3: About the presentation of Table 1.**
>
> > Your table 1 has some methods in bold even when RegretNet/CITransNet outperform them. I admit it does say in the caption that bold means “best among DSIC methods”, and the DSIC methods are separated by bars, but when reading the paper, the first thing I did was immediately look at that table, so I still found it confusing. I think it would be good to find a way to make this even more obvious.
>
> Thank you for your feedback! We will make the presentation of Table 1 more clear in the revision.

---

> > ### Comment · Reviewer_Zst9 · 2023-08-17
> > **response**
> >
> > Thanks for answering all my questions here.

---

### Author Rebuttal · Authors · 2023-08-08

We thank all reviewers for their careful comments and constructive suggestions. Here are our responses to some common questions in the reviews.

**Q1: About the comparative experiments on AMenuNet and RegretNet-based methods.**

Our primary focus in the experiments is to compare AMenuNet with other DSIC (Dominant-Strategy Incentive-Compatible) approaches, such as Lottery AMA and Item-Myerson.

While RegretNet-based methods like CITransNet and RegretFormer cover a broader range of mechanisms, they do not guarantee DSIC. These methods might achieve higher revenue levels but are at the cost of sacrificing the DSIC property.

In contrast, our DSIC approach strictly ensures that the mechanisms are DSIC, resulting in a more limited range than Regret-based methods.
Despite this narrower range, the primary advantage of AMenuNet over RegretNet-based approaches lies in its ability to ensure DSIC, prioritizing truthful bidding over revenue maximization.

Nevertheless, we still conducted the comparison experiments involving the non-DSIC RegretNet.
These experiments serve as supplementary insights rather than central facets of our study.
We do so to highlight AMenuNet's capability to achieve substantial revenue while maintaining DSIC.

**Q2: About why AMenuNet performs better than Lottery AMA.**

AMenuNet holds several advantages over Lottery AMA:
1. Handles contextual settings, adapting to diverse auction environments where bidder valuations may depend on contextual information.
2. Permutation equivariant, ensuring consistent outcomes regardless of the order of bidders or items.
3. Demonstrates generalization capabilities, performing well in auctions with varying scales and sizes.

Although AMenuNet and Lottery AMA possess the same capabilities in classical auctions given identical menu sizes, our empirical experiments demonstrate that AMenuNet excels over Lottery AMA.
Several factors potentially contribute to this superiority:
1. **Benefits of inductive bias**: While the Lottery Allocation Mechanism (Lottery AMA) calculates the allocation menu, bidder weights, and boosts as separate entities, AMenuNet takes a different approach. AMenuNet leverages an underlying neural network to compute these parameters, thereby capturing the intricate interdependencies among them. This characteristic of AMenuNet is an additional inductive bias. Adding more inductive biases usually improves the generalization ability of machine learning models [1,2].
2. **Benefits of over-parameterization**: The parameters of Lottery AMA are just the allocation menu, bidder weights, and boosts. In contrast, AMenuNet is over-parameterized with the underlying transformer-based neural network. Extensive research in deep learning has consistently demonstrated that over-parameterization offers several benefits, including improved optimization landscapes [3], better generalization [4], and less sensitivity to parameter initialization [5]. As a result, the over-parameterization of AMenuNet makes it easier to optimize than Lottery AMA, leading to better revenue performance in practice.


**References**
[1] Shalev-Shwartz, Shai, and Shai Ben-David. Understanding machine learning: From theory to algorithms. Cambridge university press, 2014.
[2] Mitchell, Tom M. "The need for biases in learning generalizations." (1980).
[3] Buhai, Rares-Darius, et al. "Empirical study of the benefits of overparameterization in learning latent variable models." International Conference on Machine Learning. PMLR, 2020.
[4] Allen-Zhu, Zeyuan, Yuanzhi Li, and Yingyu Liang. "Learning and generalization in overparameterized neural networks, going beyond two layers." Advances in neural information processing systems 32 (2019).
[5] Frankle, Jonathan, and Michael Carbin. "The lottery ticket hypothesis: Finding sparse, trainable neural networks." ICLR 2019.

---

### Decision · Program_Chairs · 2023-09-21

**Decision:**

Accept (spotlight)

**Comment:**

This paper looks at automated mechanism design through the lens of differentiable economics, roughly speaking the use of differentiable programming in economic design.  The paper looks at affine maximizer auctions (AMAs), a class of mechanism known to be representable by neural networks in a way that maintains exact DSIC and IR, but may not contain the revenue-maximizing auctions.  This is an interesting focus as most work in this space has focused on approximate DSIC with the possibility to represent the revenue-maximizing auction.  Reviewers unilaterally appreciated the experimental results showing strong revenue performance as well as the architecture that scales less dramatically with auction scale.